# FORWARD-ONLY DIFFUSION PROBABILISTIC MODELS

## ABSTRACT

This work presents a forward-only diffusion (FoD) approach for generative modelling. In contrast to traditional diffusion models that rely on a coupled forward-backward diffusion scheme, FoD directly learns data generation through a single forward diffusion process, yielding a simple yet efficient generative framework. The core of FoD is a state-dependent stochastic differential equation that involves a mean-reverting term in both the drift and diffusion functions. This mean-reversion property guarantees the convergence to clean data, naturally simulating a stochastic interpolation between source and target distributions. More importantly, FoD is analytically tractable and is trained using a simple stochastic flow matching objective, enabling a few-step non-Markov chain sampling during inference. The proposed FoD model—despite its simplicity—achieves state-of-the-art performance on various image restoration tasks. Its general applicability on image-conditioned generation is also demonstrated on diverse image-to-image translation tasks.

## 1 INTRODUCTION

The diffusion model has become a central theme in generative modelling (Sohl-Dickstein et al., 2015; Ho et al., 2020; Song et al., 2021; Karras et al., 2022). A crucial feature of the diffusion model is the use of a forward process that gradually perturbs the data into noise, coupled with a backward process that learns to transform noise back to data (Sohl-Dickstein et al., 2015; Ho et al., 2020). Benefiting from this forward-backward framework, the diffusion models have achieved remarkable performance in producing high-quality results across a wide range of applications, including image synthesis (Dhariwal & Nichol, 2021; Saharia et al., 2022b; Rombach et al., 2022; Peebles & Xie, 2023; Podell et al., 2024), translation (Meng et al., 2022; Saharia et al., 2022a; Tumanyan et al., 2023; Kawar et al., 2023; Brooks et al., 2023), and restoration (Saharia et al., 2022c; Kawar et al., 2022; Luo et al., 2023a; Wang et al., 2024; Lin et al., 2024).

Despite this success, the coupled forward-backward framework remains conceptually complex and requires learning a specialized score function, which often leads to a challenging model training problem. In addition, the necessity of corrupting data to noise in diffusion models further imposes an undesirable constraint for image-conditioned generation (Kawar et al., 2022; Saharia et al., 2022a), where the generative process ideally should start with image conditions that are structurally more informative than noise (Luo et al., 2023a; Saharia et al., 2022c; Liu et al., 2023b). This naturally leads to a fundamental question:

*"Could a conceptually simpler, single diffusion process suffice for effective generative modelling?"*

This paper answers this question affirmatively by introducing a probabilistic forward-only diffusion model (FoD). Our exploration starts from the mean-reverting stochastic differential equation (SDE) (Gillespie, 1996; Luo et al., 2023a), where the data is stochastically driven toward a specified state characterized by a fixed mean and variance. Notably, setting the mean to zero recovers the standard forward diffusion process (Song et al., 2021). Inspired by this, we propose a new form of the mean-reverting SDE, which adds mean-reversion to *both* the drift and diffusion functions as a state-dependent diffusion process. Here, we highlight the mean-reversion diffusion function as it guarantees the convergence to the noise-free mean state. By setting the mean to the target data, FoD naturally simulates the data transition between source and target distributions, without requiring a separate backward process.

We further demonstrate that the FoD process is analytically tractable and follows a multiplicative stochastic structure. Moreover, we show that the model can be learned by approximating the vector

field from each noisy state to the final clean data, a process we refer to as *stochastic flow matching*. The result is a conceptually simple, yet effective training process. Based on the tractable solution and the flow matching objective, FoD enables a few-step sampling strategy with both Markov and non-Markov chains, enabling more efficient data generation without compromising sample quality.

In addition, as a closely related work of our method, it is worth noting that flow matching (Lipman et al., 2022; Liu et al., 2022) can also eliminate the need for a separate data perturbation process by modelling a continuous flow from source distribution to target distribution. However, the noise injection, which has been shown crucial in generative models (Song & Ermon, 2019), is also eliminated due to the modelling of ordinary differential equations (ODEs). As a result, its performance drops significantly when handling image-conditioned generation tasks, such as image restoration (IR), which aims to recover high-quality images from their degraded low-quality counterparts (Albergo et al., 2023a; Martin et al., 2024; Ohayon et al., 2024). In contrast, FoD is a stochastic extension of flow matching that avoids the issue mentioned above by simulating SDEs with a state-dependent diffusion process, making it well-suited for image-conditioned generation.

Our experiments focus on image-conditioned generation, an active and fundamental direction of generative modelling with a wide range of real-world applications, including image restoration and image-to-image translation. Compared to existing diffusion and flow matching-based approaches, the proposed FoD achieves strong empirical performance across diverse tasks and datasets. Moreover, we provide a comprehensive analysis of efficient sampling using both Markov and non-Markov chains, and illustrate how the noise is injected and subsequently removed during the forward diffusion process, highlighting the importance of noise injection in image generation.

## 2 BACKGROUND

Given a source distribution $p_{\text{prior}}$ and an unknown target data distribution $p_{\text{data}}$, our goal is to build a probability path $\{p(x_t)\}_{t=0}^{T}$ that transports between the source distribution $p(x_0) = p_{\text{prior}}$ and the target distribution $p(x_T) = p_{\text{data}}$. In this paper, the source can be either noise, for unconditional generation, or images, for image-conditioned generation, e.g., image restoration.

### 2.1 DIFFUSION MODELS

Given a target data point $x_T \sim p_{\text{data}}$, diffusion models (Sohl-Dickstein et al., 2015; Ho et al., 2020) define a Markov chain forward process to progressively perturb the data into noise ($x_T \to x_0$) and then learn its reverse process to reconstruct the data ($x_0 \to x_T$). This coupled forward-backward process can be defined by stochastic differential equations (SDEs) (Song et al., 2021), given by:

$$\underbrace{\mathrm{d}x_t = f(x_t, t)\,\mathrm{d}t + g(t)\,\mathrm{d}w_t}_{\text{Forward process}} \quad \text{and} \quad \underbrace{\mathrm{d}x_t = \left[f(x_t, t) - g(t)^2\,\nabla \log p_t(x_t)\right]\mathrm{d}t + g(t)\,\mathrm{d}\bar{w}_t}_{\text{Backward process}}, \quad (1)$$

where $f(x, t)$ is the *drift* function and $g(t)$ is the *diffusion* function. Furthermore, $w$ and $\bar{w}$ are the standard Wiener process and its reverse process, respectively. We use $p_t(x_t)$ to denote the marginal probability density of $x_t$. The term $\nabla \log p_t(x_t)$, called the *score function*, is the sought-after objective in the backward (also called the reverse-time) SDE, which is often learned by a time-dependent neural network (Song et al., 2021) via score-matching. The training objective can also be converted to learn noise matching as in DDPMs (Ho et al., 2020). Moreover, the source distribution in diffusion models is often a Gaussian with a predefined mean and variance. Diffusion models typically require thousands of sampling steps to generate high-quality samples.

To enable stochastic transport between arbitrary distributions, diffusion bridge models (Zhou et al., 2024; Yue et al., 2024) introduce the Doob's $h$-transform (Doob & Doob, 1984; Särkkä & Solin, 2019) to guide the forward SDE to drift from data to a specified condition $y$:

$$\mathrm{d}x_t = f(x, t)\,\mathrm{d}t + g(t)^2 h(x_t, t, y, T) + g(t)\,\mathrm{d}w_t, \quad (2)$$

where $h(x_t, t, y, T) = \nabla_{x_t} \log p(x_T \mid x_t)\big|_{x_T = y}$ is derived from the transition density of the original forward SDE. Moreover, recent works in diffusion bridge matching (Shi et al., 2023; Liu et al., 2023a) and bridge mixtures (Peluchetti, 2023) aim to construct optimal transport paths for data generation. However, these approaches rely on either explicit bridge-consistency constraints or solving for a

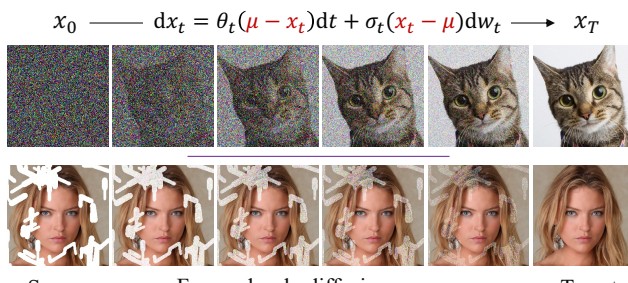

$$x_0 \longrightarrow \quad \mathrm{d}x_t = \theta_t(\mu - x_t)\mathrm{d}t + \sigma_t(x_t - \mu)\mathrm{d}w_t \quad \longrightarrow \quad x_T$$

Source ——— Forward-only diffusion process ———→ Target

Figure 1: The proposed forward-only diffusion (FoD) probabilistic model. FoD introduces the mean reversion term (marked in red color) into *both* the drift and diffusion functions, enabling high-quality data samples with a single diffusion process. This method can be easily extended from unconditional generation (top row) to image-conditioned generation, such as the image restoration in the second row.

complex mixture of diffusion bridges, which complicates both the conceptual formulation and the practical implementation. In contrast, our FoD provides a conceptually simpler and analytically tractable SDE process for effective generative modelling.

## 2.2 FLOW MATCHING GENERATIVE MODELS

Flow matching (Lipman et al., 2022; Albergo & Vanden-Eijnden, 2022; Liu et al., 2022) is a simple regression objective used for learning the velocity field $v(x_t, t)$ that transports a sample $x_t$ from the source distribution to the target distribution along the probability path $p(x_t)$ (Lipman et al., 2024). More specifically, flow matching models aim to learn the ordinary differential equation (ODE): $\mathrm{d}x_t = v(x_t, t)\,\mathrm{d}t$, where $x_0 \sim p_{\text{prior}}$ and the drift $v(x_t, t)$ transports samples from $x_0$ to $x_1 \sim p_{\text{data}}$. Here, each latent variable $x_t$ in the ODE path is drawn by linearly interpolating source and target data samples, i.e., $x_t = tx_1 + (1 - t)x_0$. Then the training can be performed by uniformly sampling data pairs and timesteps and optimizing a flow matching objective, as:

$$L_{\text{FM}}(\phi) = \mathbb{E}_{x_0, x_1, t \sim \mathcal{U}(0,1)}\big[\|(x_1 - x_0) - v_\phi(x_t, t)\|^2\big], \tag{3}$$

where $v_\phi(x_t, t)$ is a neural network approximating the true velocity field. Flow matching models eliminate the diffusion term from the generative process and thus lead to a simpler and more direct learning procedure based on ODE paths. However, in this paper, we observe that applying it to image-conditioned generation tasks, such as image restoration, leads to a significant performance drop due to the lack of stochastic noise injection (see Section 4.1 for more details). Moreover, it is worth noting that both diffusion models and flow matching models can be unified into the stochastic interpolants (Albergo et al., 2023a) framework.

## 3 FORWARD-ONLY DIFFUSION PROBABILISTIC MODELS

The key value of the forward-only diffusion (FoD) model lies in defining an analytically solvable, forward-only process that removes the need to approximate or learn a reverse SDE. This makes the generative process conceptually simple and stable, and easier to extend to image-conditioned generation, as illustrated in Figure 1.

### 3.1 PRELIMINARIES: MEAN-REVERTING SDE

Our exploration starts from a mean-reverting SDE (Gillespie, 1996; Luo et al., 2023a) where the data is stochastically driven towards a state characterized by a specified mean $\mu$ and variance $\lambda^2$:

$$\mathrm{d}x_t = \theta_t(\mu - x_t)\,\mathrm{d}t + \sigma_t\,\mathrm{d}w_t, \tag{4}$$

where $\{\theta_t\}_{t=0}^T$ and $\{\sigma_t\}_{t=0}^T$ are positive mean-reversion and diffusion schedules, respectively. By coupling the schedules as $\sigma_t^2 / \theta_t = 2\lambda^2$ for all $t$, we obtain the following solution (Luo et al., 2023a)

$$x_t = \mu + (x_0 - \mu)\,e^{-\int_0^t \theta_z\,\mathrm{d}z} + \int_0^t \sigma_z\,e^{-\int_s^t \theta_s\,\mathrm{d}s}\,\mathrm{d}w_z. \tag{5}$$

As $t \to \infty$, the SDE converges to a stationary state $x_T \sim \mathcal{N}(x_T \mid \mu, \lambda^2)$. This property suggests constructing a process that transports samples from the source distribution $p_{\text{prior}}$ to the target distribution $p_{\text{data}}$, by setting the mean $\mu$ to be a sample from $p_{\text{data}}$. However, as can be observed from Eq. (5), the resulting sample $x_T$ is still noisy, with variance $\lambda^2$, which works against our goal of generating high-quality clean data samples. In the following sections, we address this problem by introducing mean-reversion in both the drift *and* diffusion functions.

## 3.2 Forward-only Diffusion Process

We begin by designing an SDE with mean-reversion terms in both the drift and diffusion functions, as

$$\mathrm{d}x_t = \theta_t\,(\mu - x_t)\,\mathrm{d}t + \sigma_t\,(x_t - \mu)\,\mathrm{d}w_t. \tag{6}$$

This is a state-dependent linear SDE with multiplicative noise, where the diffusion volatility increases in the beginning steps and then decreases to zero when $x_t$ converges to $\mu$. We typically use $x_t - \mu$ in the diffusion function such that this SDE simulates a reverse Wiener process as in diffusion models (Song et al., 2021). For image generation, the noise $\mathrm{d}w_t$ is added independently at each pixel, meaning that this SDE is applied for image transitions pixel-by-pixel, under the Itô interpretation.

We refer to this SDE as the *forward-only diffusion* (FoD) process, and present its solution as follows:

**Proposition 3.1.** *Given an initial state $x_s$ at time $s < t$, the unique solution to the SDE (6) is*

$$x_t = \left(x_s - \mu\right)\mathrm{e}^{-\int_s^t \left(\theta_z + \frac{1}{2}\sigma_z^2\right)\mathrm{d}z + \int_s^t \sigma_z \mathrm{d}w_z} + \mu, \tag{7}$$

*where the stochastic integral is interpreted in the Itô sense and can be reparameterised as $\bar{\sigma}_{s:t}\,\epsilon$, where $\bar{\sigma}_{s:t} = \sqrt{\int_s^t \sigma_z^2\,\mathrm{d}z}$, and $\epsilon \sim \mathcal{N}(0, I)$ is a standard Gaussian noise.*

The proof is provided in Appendix A.1. The use of positive $\theta$ and $\sigma$ schedules in Eq. (7) introduces a strong exponential damping factor that drives $x_t$ toward $\mu$, thereby stabilizing the process over time. In addition, the solution in Eq. (7) shows that the stochastic flow field $\mu - x_t$ forms a Geometric Brownian motion (Ross, 2014) and yields the following corollary:

**Corollary 3.2.** *Under the same assumptions as in Proposition 3.1, the stochastic flow field $\mu - x_t$ satisfies the multiplicative stochastic structure. More precisely, it is log-normally distributed by*

$$\log(\mu - x_t) \sim \mathcal{N}\left(\log(\mu - x_s) - \int_s^t \left(\theta_z + \frac{1}{2}\sigma_z^2\right)\mathrm{d}z, \int_s^t \sigma_z^2\,\mathrm{d}z\,I\right). \tag{8}$$

This follows directly from Proposition 3.1, by rearranging $\mu - x_t$ to the left of Eq. (7) and applying the logarithm to both sides (see Appendix A.1). The subtractive form of the logarithm reflects that the flow field decays multiplicatively from its initial value with a stochastic exponential scaling.

*Notational Clarifications:* Although the sign of $\mu - x_t$ in general can be either positive or negative, it remains consistent across all times $t$ for a given sample; therefore, we choose to omit absolute values inside the logarithmic terms in Eq. (8) for notational convenience. In addition, we further let $\bar{m}_{s:t} = -\int_s^t (\theta_z + \frac{1}{2}\sigma_z^2)\,\mathrm{d}z$ and $\bar{m}_t = \bar{m}_{0:t}$ in the rest of the paper to simplify the notation.

## 3.3 Stochastic Flow Matching

Let us now explain how we can learn this FoD process, i.e., transforming data from a known source distribution $p_{\text{prior}}$ to an unknown target distribution $p_{\text{data}}$. Following DDPMs (Ho et al., 2020), we define the FoD model as $p_\phi(x_{0:T})$, a joint distribution with learnable transitions starting at $x_0$, as

$$p_\phi(x_{0:T}) = p_{\text{prior}}(x_0)\prod_{t=0}^{T-1} p_\phi(x_{t+1} \mid x_t), \qquad x_0 \sim p_{\text{prior}}. \tag{9}$$

We propose to set the transition kernel $p_\phi(x_{t+1}|x_t)$ to be in the same log-Gaussian form as Eq. (7). The training can then be performed by minimizing the negative log-likelihood of $p_\phi(x_T)$, which is equivalent to optimizing the following objective:

$$\mathbb{E}_p\left[\sum_{t=0}^{T-1} D_{KL}(p(x_{t+1} \mid x_t, x_T) \,\|\, p_\phi(x_{t+1} \mid x_t))\right]. \tag{10}$$

The proof is provided in Appendix A.2. During training, we set $x_T$ equal to $\mu$ such that the SDE (6) converges to data $\mu$ exactly. Then, the conditional distribution $p(x_{t+1}|x_t, x_T = \mu)$ is tractable as shown in Eq. (7). By letting the functions $f_\mu(x_t) = \mu - x_t$ and $f_\phi(x_t, t) = \hat{\mu}_\phi - x_t$ denote the ground truth and the model prediction of the stochastic flow field, respectively, we transform the distributions in Eq. (10) from SDE states to stochastic flow fields. Note that this transformation, i.e.,

---

**Algorithm 1** FoD Training

**Require:** $p_{\text{prior}}$, $p_{\text{data}}$, model $f_\phi$
1: **repeat**
2:    $x_0 \sim p_{\text{prior}}$, $\mu \sim p_{\text{data}}$
3:    $\epsilon \sim \mathcal{N}(0, I)$, $t \sim \text{Uniform}(\{1, \ldots, T\})$
4:    $x_t = (x_0 - \mu)\, e^{\bar{m}_t + \sigma_t\, \epsilon} + \mu$
5:    Take gradient descent step on
         $\nabla_\phi \| (\mu - x_t) - f_\phi(x_t, t) \|^2$
6: **until** converged

---

**Algorithm 2** FoD Sampling

**Require:** $p_{\text{prior}}$, time interval $\Delta t$, model $f_\phi$
1: $x_0 \sim p_{\text{prior}}$
2: **for** $t = 0, \ldots, T - 1$ **do**
3:    $\epsilon \sim \mathcal{N}(0, I)$
4:    $\Delta x = \theta_t f_\phi(x_t, t) \cdot \Delta t - \sigma_t f_\phi(x_t, t) \cdot \sqrt{\Delta t}\, \epsilon$
5:    $x_{t+1} = x_t + \Delta x$
6: **end for**
7: **return** $x_T$

---

**Algorithm 3** Markov Chain Sampling

**Require:** $p_{\text{prior}}$, step size $k$, model $f_\phi$
1: $x_0 \sim p_{\text{prior}}$
2: **for** $t = 0, k, 2k, \ldots, T$ **do**
3:    $\epsilon \sim \mathcal{N}(0, I)$
4:    $\hat{\mu} = x_t + f_\phi(x_t, t)$
5:    $x_{t+k} = \left( \boxed{x_t} - \hat{\mu} \right) e^{\bar{m}_{t:t+k} + \epsilon \cdot \bar{\sigma}_{t:t+k}} + \hat{\mu}$
6: **end for**
7: **return** $x_T$

---

**Algorithm 4** Non-Markov Chain Sampling

**Require:** $p_{\text{prior}}$, step size $k$, model $f_\phi$
1: $x_0 \sim p_{\text{prior}}$
2: **for** $t = 0, k, 2k, \ldots, T$ **do**
3:    $\epsilon \sim \mathcal{N}(0, I)$
4:    $\hat{\mu} = x_t + f_\phi(x_t, t)$
5:    $x_{t+k} = \left( \boxed{x_0} - \hat{\mu} \right) e^{\bar{m}_{t+k} + \epsilon \cdot \bar{\sigma}_{t+k}} + \hat{\mu}$
6: **end for**
7: **return** $x_T$

---

from $p(\cdot | x_t, \mu)$ to $p(\cdot | f_\mu(x_t))$, holds because its Jacobian determinant equals one. Instead of Eq. (10), we can therefore minimize the KL divergence between two stochastic flow distributions:

$$\mathbb{E}_p \Big[ \sum_{t=0}^{T-1} D_{KL}(p(f_\mu(x_{t+1}) \mid f_\mu(x_t) \,\|\, p(f_\phi(x_{t+1}, t) \mid f_\phi(x_t, t))) \Big]. \tag{11}$$

Combining this with Corollary 3.2, we obtain the final objective:

$$
\begin{aligned}
L_{\text{SFM}}(\phi) &:= \mathbb{E}_{\mu \sim p_{\text{data}}, x_t \sim p(x_t | x_0, \mu)} \Big[ \| \log(\mu - x_t) - \log f_\phi(x_t, t) \|^2 \Big] \\
&\approx \mathbb{E}_{\mu \sim p_{\text{data}}, x_t \sim p(x_t | x_0, \mu)} \Big[ \| (\mu - x_t) - f_\phi(x_t, t) \|^2 \Big],
\end{aligned}
\tag{12}
$$

where the approximation follows from a first-order Taylor expansion close to the optimum. Please refer to Appendix A.3 for more details. This objective is referred to as *stochastic flow matching* and it is entirely linear, which leads to a simple and numerically stable training process.

The standard training and sampling (via the Euler–Maruyama method) procedures are provided in Algorithm 1 and Algorithm 2, respectively. In addition, the target data estimate $\hat{\mu}$ is given by

$$\hat{\mu} = x_t + f_\phi(x_t, t), \tag{13}$$

which can be applied to the forward transition (7) for fast data sampling.

**Fast Sampling with Markov and non-Markov Chains** While the generation can be performed by iteratively solving the SDE (6) with numerical schemes such as the Euler–Maruyama method, it often requires hundreds of sampling steps. Fortunately, the tractable solution of FoD naturally enables fast sampling during inference, by choosing times discretely with a larger step size $k$, as $t = [0, k, 2k, 3k, \ldots, T]$, where $T$ is the total number of timesteps. Since our prediction at each step is the sought-after target data $\hat{\mu}$, the next state $x_{t+k}$ can be sampled following Eq. (7) with either Markov or non-Markov chains. This is done by setting the transition to $x_t \to x_{t+k}$ or $x_0 \to x_{t+k}$, as illustrated in Algorithm 3 and Algorithm 4, respectively. A further discussion is provided in Section 5.

### 3.4 Connection to Prior Work

In this section, we establish the theoretical connections between FoD and two closely related prior works: stochastic interpolants (SI) (Albergo et al., 2023a) and flow matching (FM) (Lipman et al., 2022). SI provides a unified stochastic framework to bridge two arbitrary distributions, while FM formulates a deterministic transport map between two distributions via an ODE.

**Stochastic Interpolants**  Let us recall the solution in Eq. (7) of the FoD process. By setting the initial state to $x_0$ and rearranging the equation, we obtain a stochastic process in the interpolant form:

$$x_t = I(t, x_0, \mu) = x_0 \, \alpha_t + \mu \, (1 - \alpha_t), \quad \alpha_t = \mathrm{e}^{-\int_0^t \left(\theta_z + \frac{1}{2}\sigma_z^2\right) \mathrm{d}z + \int_0^t \sigma_z \mathrm{d}w_z}. \tag{14}$$

Here, $I(t, x_0, \mu)$ satisfies the boundary conditions of a stochastic interpolant, with randomness introduced via $\mathrm{d}w$. FoD can thus be viewed as a powerful instantiation of SI, distinguished by two key properties: multiplicative log-normal interpolation and a state-dependent stochastic path from $x_0$ to $\mu$. This formulation allows noise to be gradually added and subsequently removed within a single forward process. This perspective helps unify FoD with a broader class of generative frameworks.

**Flow Matching**  We consider a deterministic version of the FoD process in Eq. (6), i.e., omitting the diffusion term or setting $\sigma_t = 0$ for all times. This gives a mean-reverting ODE that bridges two distributions without noise injection, as $\mathrm{d}x_t = \theta_t \, (\mu - x_t) \, \mathrm{d}t$ with solution $x_t = (x_s - \mu) \, \mathrm{e}^{-\int_s^t \theta_z \, \mathrm{d}z} + \mu$. Setting $s$ to 0 and rewriting this solution yields an interpolation between $x_0$ and $\mu$:

$$x_t = x_0 \, \alpha_t + \mu \, (1 - \alpha_t), \quad \alpha_t = \mathrm{e}^{-\int_0^t \theta_z \, \mathrm{d}z}, \tag{15}$$

which forms a similar transportation path as in flow matching but with a special velocity field given by $\theta_t \, (\mu - x_t)$. We can then learn the drift, resulting in a conditional flow matching objective:

$$L_{\mathrm{CFM}} \coloneqq \mathbb{E}_{\mu, x_t} \left[ \|(\mu - x_t) - f_\phi(x_t, t)\|^2 \right] = \mathbb{E}_{\mu, x_t} \left[ \|\alpha_t(\mu - x_0) - f_\phi(x_t, t)\|^2 \right], \tag{16}$$

which is a deterministic form of the stochastic flow matching in Eq. (12). In practice, we can learn the target displacement $\mu - x_0$ directly and define the $\alpha$ schedule to be linear, i.e., decreasing from 1 to 0, in which case this mean-reverting ODE becomes flow matching with a straight-line path (Lipman et al., 2022; Liu et al., 2022) exactly. In other words, our primary FoD model can also be regarded as a stochastic extension of flow matching models.

## 4 EXPERIMENTS

Our experiments mainly focus on image restoration (IR), a fundamental problem in computer vision which aims to accurately recover high-quality images from their degraded low-quality counterparts. The general applicability of our FoD model on image-conditioned generation is further demonstrated via diverse image-to-image translation tasks[1].

**Implementation and Setup**  We use a U-Net (Ronneberger et al., 2015) architecture similar to DDPM (Ho et al., 2020) for flow prediction in all tasks. Attention layers are removed for efficient training and testing, similar to IR-SDE (Luo et al., 2023a;b). We choose the commonly used cosine and linear schedules (Nichol & Dhariwal, 2021) for $\theta_t$ and $\sigma_t$, respectively, and normalize $\sigma_t^2$ to sum to 1 to ensure numerical stability under multiplicative noise perturbation. The number of sampling steps is fixed to 100 for all tasks. We use the AdamW (Loshchilov & Hutter, 2017) optimizer with parameters $\beta_1 = 0.9$ and $\beta_2 = 0.99$. The training requires $500\,000$ iterations with a learning rate of $10^{-4}$. All our models are trained on an A100 GPU with 40 GB of memory for about 1.5 days.

The Learned Perceptual Image Patch Similarity (LPIPS) (Zhang et al., 2018) and Fréchet Inception Distance (FID) (Heusel et al., 2017) are reported to evaluate the perceptual fidelity and overall visual quality. Additionally, Peak Signal-to-Noise Ratio (PSNR) and Structural Similarity Index Measure (SSIM) (Wang et al., 2004) are also included to evaluate pixel-level and structural similarity.

### 4.1 IMAGE RESTORATION

We evaluate our method on four IR tasks: 1) image deraining on the Rain100H dataset (Yang et al., 2017), 2) dehazing on the RESIDE-6k dataset (Qin et al., 2020), 3) low-light enhancement on LOL (Wei et al., 2018), and 4) face inpainting on CelebA-HQ (Karras et al., 2017).

In our experiments, we select IR-SDE (Luo et al., 2023a) as the main comparison method to evaluate the performance gap between forward-backward and forward-only schemes for diffusion-based

---

[1]We provide more training details, datasets, and unconditional generation results in the Appendix.

Table 1: Comparison of FoD with other diffusion, diffusion bridge, and flow matching approaches on four different image restoration datasets, evaluated using both distortion and perceptual metrics.

| Method | Distortion | | Perceptual | |
|---|---|---|---|---|
| | PSNR↑ | SSIM↑ | LPIPS↓ | FID↓ |
| U-Net baseline | 29.12 | 0.882 | 0.153 | 57.55 |
| IR-SDE | 31.65 | 0.904 | 0.047 | 18.64 |
| GOUB | 31.96 | 0.903 | 0.046 | 18.14 |
| UniDB | 32.05 | 0.904 | 0.045 | 17.65 |
| ReFlow | 28.36 | 0.871 | 0.152 | 64.81 |
| PMRF | 29.01 | 0.857 | 0.173 | 69.25 |
| FoD (Ours) | **32.56** | **0.925** | **0.038** | **14.10** |

(a) Deraining results on the Rain100H dataset.

| Method | Distortion | | Perceptual | |
|---|---|---|---|---|
| | PSNR↑ | SSIM↑ | LPIPS↓ | FID↓ |
| U-Net baseline | 20.51 | 0.808 | 0.162 | 75.84 |
| IR-SDE | 20.45 | 0.787 | 0.129 | 47.28 |
| GOUB | 19.29 | 0.775 | 0.148 | 50.44 |
| UniDB | 20.18 | 0.796 | 0.128 | 45.61 |
| ReFlow | 19.62 | 0.767 | 0.221 | 91.93 |
| PMRF | 19.32 | 0.753 | 0.189 | 81.59 |
| FoD (Ours) | **21.61** | **0.819** | **0.105** | **41.31** |

(b) Low-light enhancement on the LOL dataset.

| Method | Distortion | | Perceptual | |
|---|---|---|---|---|
| | PSNR↑ | SSIM↑ | LPIPS↓ | FID↓ |
| U-Net baseline | 22.88 | 0.906 | 0.065 | 15.65 |
| IR-SDE | 25.25 | 0.906 | 0.060 | 8.33 |
| GOUB | 25.31 | 0.908 | 0.048 | 8.21 |
| UniDB | 25.65 | 0.896 | 0.051 | 8.29 |
| ReFlow | 20.84 | 0.864 | 0.081 | 23.53 |
| PMRF | 22.45 | 0.868 | 0.092 | 24.09 |
| FoD (Ours) | **26.57** | **0.932** | **0.033** | **8.14** |

(c) Dehazing results on the RESIDE-6k dataset.

| Method | Distortion | | Perceptual | |
|---|---|---|---|---|
| | PSNR↑ | SSIM↑ | LPIPS↓ | FID↓ |
| U-Net baseline | 27.97 | 0.889 | 0.097 | 58.78 |
| IR-SDE | 29.83 | 0.904 | 0.045 | 26.30 |
| GOUB | 29.81 | 0.916 | 0.039 | 23.39 |
| UniDB | 30.01 | 0.917 | 0.038 | 23.16 |
| ReFlow | 29.84 | 0.912 | 0.065 | 38.65 |
| PMRF | **30.45** | 0.901 | 0.082 | 55.40 |
| FoD (Ours) | 30.28 | **0.923** | **0.029** | **16.12** |

(d) Inpainting results on the CelebA-HQ dataset.

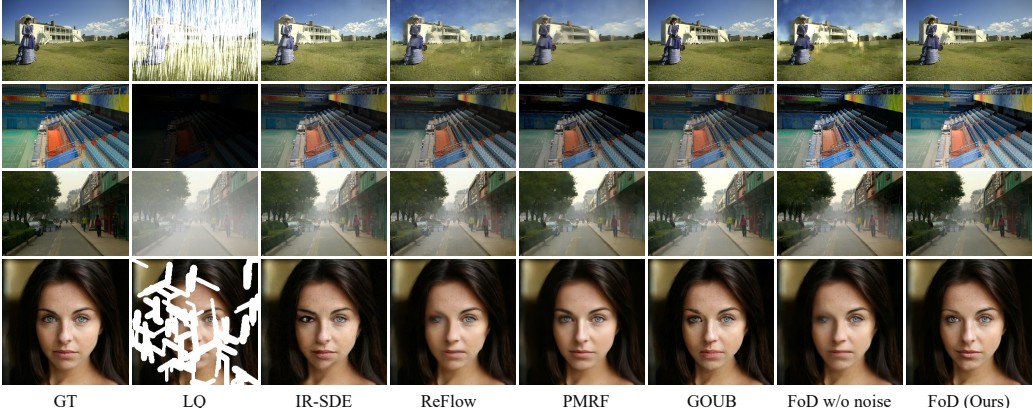

GT · LQ · IR-SDE · ReFlow · PMRF · GOUB · FoD w/o noise · FoD (Ours)

Figure 2: Comparison of FoD with other approaches on four IR tasks. Here, we add one column for the results of 'FoD w/o noise' to illustrate the importance of noise injection in image restoration.

restoration. We further compare against two state-of-the-art diffusion bridge models, GOUB (Yue et al., 2024) and UniDB (Zhu et al., 2025), both of which leverage mean-reverting bridges for image conditioned generation. Moreover, we implement two flow matching based approaches, Rectified flow (Liu et al., 2022; Liu, 2022) and posterior-mean rectified flow (PMRF) (Ohayon et al., 2024), that learn ODEs and also enable the model to generate images with a single forward process. In addition, a U-Net model, using the same architecture as our FoD, is trained with the $\ell_1$ loss as a CNN baseline on all tasks for reference.

The quantitative comparisons on four IR tasks are reported in Table 1. The proposed FoD achieves the best results across all datasets in comparison to other diffusion, diffusion bridge, and flow-based approaches. Compared to the U-Net baseline, IR-SDE successfully improves the results on perceptual metrics (LPIPS and FID), proving the effectiveness of the forward-backward based diffusion IR schemes. By leveraging bridges, GOUB and UniDB further improve the IR-SDE results across most tasks and metrics, but are still consistently outperformed by our FoD. We also observe that flow-based approaches, such as ReFlow and PMRF, perform inferiorly across all metrics. While PMRF improves the PSNR results for flow matching-based IR, the performance gain potentially comes from the two-stage training strategy and the small noise injection in the initial state of rectified flow.

Table 2: Results of different sampling approaches using the same trained FoD model. Here, 'FoD w/ EM' denotes the *100-step* Euler–Maruyama sampling method, while 'FoD w/ MC' and 'FoD w/ NMC' denote *10-step* Markov and non-Markov chain fast sampling, respectively.

| Method | Deraining | | | Low-light enhance | | | Dehazing | | | Inpainting | | |
|---|---|---|---|---|---|---|---|---|---|---|---|---|
| | PSNR↑ | LPIPS↓ | FID↓ | PSNR↑ | LPIPS↓ | FID↓ | PSNR↑ | LPIPS↓ | FID↓ | PSNR↑ | LPIPS↓ | FID↓ |
| FoD w/ EM | 32.56 | 0.038 | 14.10 | 21.61 | 0.105 | 41.31 | 26.57 | 0.033 | 8.14 | 30.28 | 0.029 | 16.12 |
| FoD w/ MC | 33.27 | 0.039 | 15.14 | 23.12 | 0.093 | 32.37 | 26.76 | 0.031 | 10.07 | 31.02 | 0.031 | 18.06 |
| FoD w/ NMC | 33.63 | 0.041 | 15.64 | 23.05 | 0.098 | 47.87 | 26.77 | 0.032 | 10.31 | 31.32 | 0.038 | 23.28 |

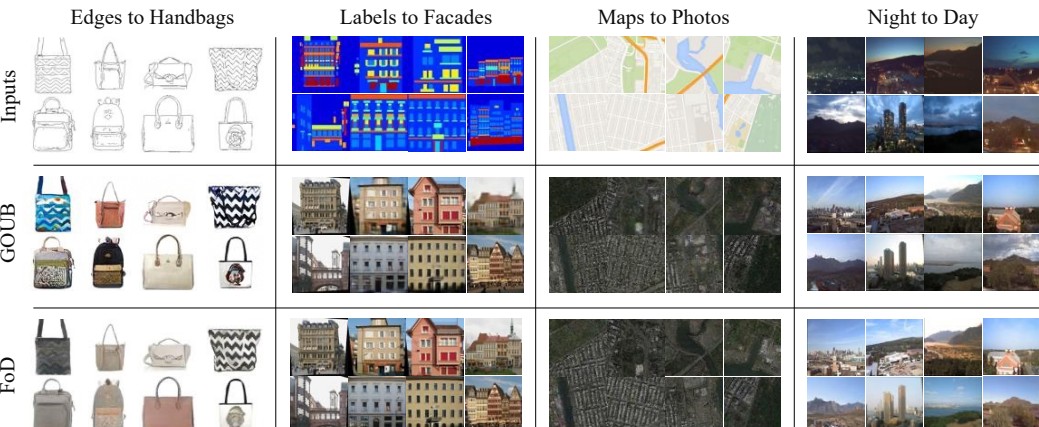

Figure 3: Qualitative results of FoD and GOUB on diverse image-to-image translation tasks.

We also provide visual comparisons in Figure 2, showing that our FoD produces the most realistic and high-fidelity results. In particular, while all deterministic approaches without noise injection (ReFlow, PMRF and 'FoD w/o noise') tend to generate overly smooth outputs (see e.g. the left eye area in the face inpainting case), the proposed FoD model consistently produces sharper and more detailed images. Further discussion on the role of noise injection is provided in Section 5.

## 4.2 IMAGE-TO-IMAGE TRANSLATION

We also perform experiments on diverse image-to-image translation tasks, to further demonstrate the general applicability of the proposed FoD method in image-conditioned generation. The main tasks include edges to handbags (E2H) (Isola et al., 2017), labels to facades (L2F) (Tyleček & Šára, 2013), maps to aerial photos (M2A) (Isola et al., 2017), and night to day (N2D) (Laffont et al., 2014). All images of the four tasks are resized to $64 \times 64$ resolution. We adopt the same implementa-

Table 3: Comparison of FID results on four image-to-image translation tasks.

| Method | E2H | L2F | M2A | N2D |
|---|---|---|---|---|
| Flow matching | 25.29 | 21.30 | 36.07 | 78.52 |
| GOUB | 9.36 | 3.47 | 11.62 | 64.35 |
| UniDB | 9.12 | 2.79 | 11.78 | 64.52 |
| FoD (Ours) | **8.45** | **0.86** | **2.33** | **52.11** |

tion as in the image restoration setting, except that the fast sampling method, FoD with 10-step MC, is used for all tasks. For comparison, we implement a flow matching (Lipman et al., 2022) method and two diffusion bridge models (GOUB (Yue et al., 2024) and UniDB (Zhu et al., 2025)) similar to the image restoration experiments. Following the evaluation setup of previous works (Li et al., 2023; Zhou et al., 2024), we report the FID results on the training datasets of different tasks in Table 3. We observe that flow matching achieves significantly worse performance than the other approaches, which all perform noise injection in the data sampling. A more comprehensive evaluation (with additional MSE, LPIPS, and FID metrics) on these tasks are provided in Appendix C.4 (Table 8). Our FoD achieves the best performance across all datasets and metrics, further illustrating its generality in conditional image generation. Qualitative comparisons across these different tasks are shown in Figure 3. One can observe that, while both FoD and GOUB are capable of handling complex image-to-image translation problems, FoD produces sharper and more reliable results when the source and target domains differ substantially, as seen in the bottom-left example of *edges to handbags* and in most cases of *maps to aerial photos*. Notably, the *night to day* dataset involves many-to-many mappings due to temporal variability, yet FoD still produces satisfactory results, demonstrating its strong generative capability.

Table 4: Quantitative results of FoD and its noise-free variant on four image restoration tasks.

| Method | Deraining | | | Low-light enhance | | | Dehazing | | | Inpainting | | |
|--------|-----------|--|--|-------------------|--|--|----------|--|--|------------|--|--|
| | PSNR↑ | LPIPS↓ | FID↓ | PSNR↑ | LPIPS↓ | FID↓ | PSNR↑ | LPIPS↓ | FID↓ | PSNR↑ | LPIPS↓ | FID↓ |
| FoD w/o noise | 29.44 | 0.132 | 59.54 | 19.96 | 0.193 | 86.21 | 24.18 | 0.048 | 13.80 | 29.94 | 0.065 | 38.78 |
| FoD (Ours) | 32.56 | 0.038 | 14.10 | 21.61 | 0.105 | 41.31 | 26.57 | 0.033 | 8.14 | 30.28 | 0.029 | 16.12 |

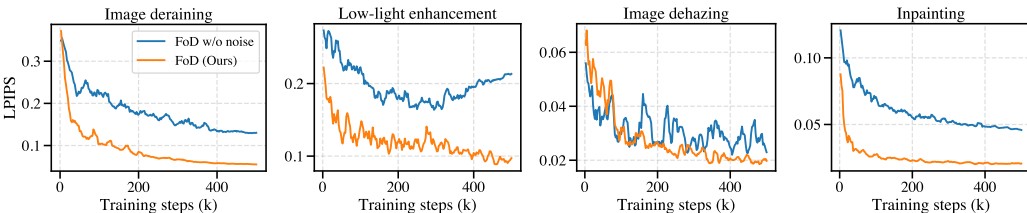

Figure 4: Training curves of FoD and its noise-free variant on four image restoration tasks.

## 5 DISCUSSION

**Fast Sampling** As discussed in Section 3.3, FoD naturally supports fast sampling using both Markov and non-Markov chains. Table 2 demonstrates that with only 10 sampling steps, the non-Markov variant achieves even better results than the standard Euler–Maruyama solver in terms of PSNR, without significantly compromising perceptual quality. To investigate the impact of sampling steps, we provide a detailed analysis on image restoration in Figure 5. For comparison, the Euler–Maruyama solver with 100 steps is included as a baseline. We observe that both sampling strategies yield improved distortion metrics as the number of steps *decreases*, especially in the low-step areas (5–20), where they substantially outperform the baseline with a small perceptual performance drop. Similar trends are observed for LPIPS and FID, although both metrics show a modest decline in performance when reducing the number of steps from 20 to 5. Note that FoD with Markov chains applies the FoD transition recursively at each step. The prediction error thus accumulates over time, which distorts spatial structure (lower SSIM) but increases sample diversity and quality (better FID). In contrast, FoD with non-Markov chains preserves the structure since all its transitions start from the initial state. More comparisons and illustrations of the forward process are provided in Appendix C.2. In practice, the MC sampler is preferable for highly ill-posed problems, while the non-MC sampler is more suitable for structure-preserving tasks such as image restoration.

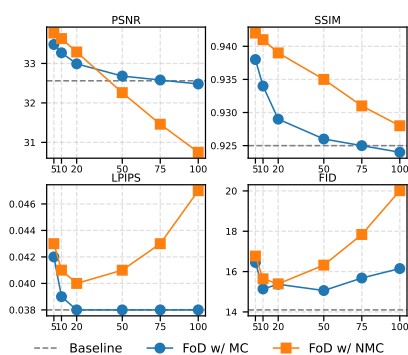

Figure 5: Comparison of fast sampling with Markov chain (MC) vs. with non-Markov chain (NMC) on the deraining task, using the same pretrained model.

**Effectiveness of Noise Injection** This section explores the importance of noise injection in image-conditioned generation. To this end, we conduct an additional experiment where a noise-free variant of FoD is trained on four image restoration tasks, by setting $\sigma_t = 0$ for all $t$. This effectively reduces FoD to a flow matching model, as described in Section 3.4. We keep the $\theta$ schedule the same as FoD for a fair comparison. Quantitative results in Table 4 show a significant drop in performance across all tasks and metrics. The corresponding visual results are also provided in the second-to-last column of Figure 2, where the produced images are blurry and unclear compared to those of our original FoD. Moreover, the training curves of FoD with and without noise injection on different image restoration tasks are provided in Figure 4. FoD consistently outperforms its noise-free variant on all tasks, further demonstrating the effectiveness and importance of noise injection in image-conditioned generation.

**FoD Sampling Process** We compare the sampling processes of FoD and the GOUB diffusion bridge model on a face inpainting example in Figure 6 (left). Both methods stochastically interpolate between the two distributions, but while the diffusion bridge adds noise across the entire image, our FoD focuses only on the degraded areas. Furthermore, diffusion bridges still rely on the coupled forward-backward framework, whereas FoD employs a simple, single forward diffusion process.

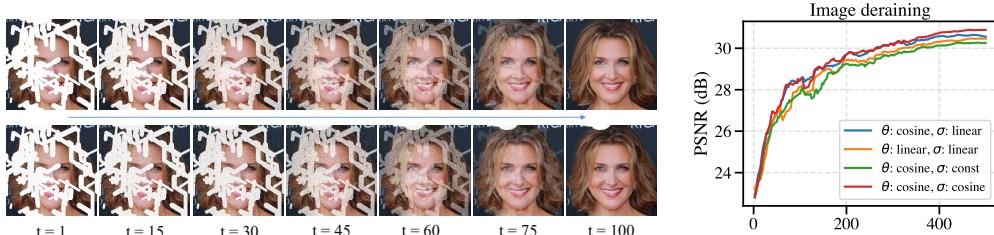

Figure 6: **Left:** Comparison of the sampling process between FoD (top) and diffusion bridges (bottom) on face inpainting. **Right:** Training curves of different $\theta$ and $\sigma$ schedules on image deraining.

**Choice of $\theta$ and $\sigma$ Schedules** The choice of noise schedules can be curcial for image generation performance (Nichol & Dhariwal, 2021; Chen, 2023). To examine the sensitivity of our FoD model to $\theta$ and $\sigma$ schedules, we conduct an ablation study in which various schedule combinations are adopted for the image deraining task, as illustrated in the right of Figure 6 and in Appendix C.2 (Table 6 and Figure 10). In general, FoD offers a flexible framework that accommodates different $\sigma$ schedules, while adopting the cosine schedule for both parameters yields the best performance.

**Limitations and Future Work** Although FoD performs well for image-conditioned generation, its multiplicative structure in the forward process poses a challenge for unconditional generation, where the source distribution is typically Gaussian. Specifically, injecting log-Gaussian noise (as defined in Eq. (7)) into a source sample $x_0 \sim \mathcal{N}(0, I)$ complicates the learning process and leads to a decline in sample quality (See Appendix C.3 for additional details). In future work, we plan to explore log-space transformations and optimal transport-based drift paths to improve the unconditional generation capabilities, and further extend the model to complex text-to-image generation tasks.

## 6 RELATED WORK

Denoising diffusion models (Sohl-Dickstein et al., 2015; Ho et al., 2020; Song et al., 2021; Karras et al., 2022; Kim et al., 2025) and flow matching models (Lipman et al., 2022; Liu et al., 2022; Lipman et al., 2024; Albergo et al., 2023a; Gat et al., 2024) are two popular frameworks in generative modelling and have been widely applied to various applications including image generation (Dhariwal & Nichol, 2021; Ho et al., 2022; Peebles & Xie, 2023; Ho & Salimans, 2022; Ma et al., 2024), text-to-image generation (Rombach et al., 2022; Ruiz et al., 2023; Podell et al., 2024; Saharia et al., 2022b), image translation (Lugmayr et al., 2022; Saharia et al., 2022a; Su et al., 2022; Xia et al., 2024b; Liu et al., 2023b; Ben-Hamu et al., 2024; Li et al., 2023; Xia et al., 2024a; Zheng et al., 2024), etc. Inspired by their success in producing photo-realistic images conforming to human preference, these models have recently been applied to image restoration for advanced performance (Wang et al., 2024; Saharia et al., 2022c; Yue et al., 2023; Kawar et al., 2022; Yue et al., 2024; Luo et al., 2024a;b; Liu et al., 2024; Shi et al., 2024). In addition, Albergo et al. (2023a) unify diffusion and flow matching models through stochastic interpolants. This formulation has also been applied to image restoration (Albergo et al., 2023b), where stochastic flows guided by corrupted observations recover clean images, effectively serving as a stochastic extension of flow matching for inverse problems. Subsequent works enhance this approach using pretrained flow matching models (Ben-Hamu et al., 2024) or refined training pipelines (Ohayon et al., 2024). These works are closely related to ours, but they all adopt noise-free generation processes. In contrast, FoD involves a state-dependent diffusion process for image generation, which is naturally well-suited for image-conditioned generation, and which degrades to flow matching when the diffusion term vanishes.

## 7 CONCLUSION

This paper presents a new framework, named FoD, for generative modelling with a single forward diffusion process. We show that FoD is analytically tractable and can be trained using a simple stochastic flow matching objective. Our model is evaluated on various image-conditioned generation tasks, including image restoration and image-to-image translation. FoD achieves strong performance compared to other diffusion models and flow matching approaches, demonstrating its effectiveness and efficiency in generative modelling, particularly for image restoration.

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

# A PROOFS

## A.1 PROOF TO PROPOSITION 3.1 AND COROLLARY 3.2

**Proposition 3.1.** *Given an initial state $x_s$ at time $s < t$, the unique solution to the SDE (6) is*

$$x_t = (x_s - \mu) \, \mathrm{e}^{- \int_s^t \left(\theta_z + \frac{1}{2}\sigma_z^2\right) \mathrm{d}z + \int_s^t \sigma_z \mathrm{d}w_z} + \mu, \tag{17}$$

*where the stochastic integral is interpreted in the Itô sense and can be reparameterised as $\bar{\sigma}_{s:t} \, \epsilon$, where $\bar{\sigma}_{s:t} = \sqrt{\int_s^t \sigma_z^2 \, \mathrm{d}z}$, and $\epsilon \sim \mathcal{N}(0, I)$ is a standard Gaussian noise.*

**Corollary 3.2.** *Under the same assumptions as in Proposition 3.1, the stochastic flow field $\mu - x_t$ satisfies the multiplicative stochastic structure. More precisely, it is log-normally distributed by*

$$\log(\mu - x_t) \sim \mathcal{N}\left(\log(\mu - x_s) - \int_s^t \left(\theta_z + \frac{1}{2}\sigma_z^2\right) \mathrm{d}z, \ \int_s^t \sigma_z^2 \, \mathrm{d}z \, I\right). \tag{18}$$

*Proof.* Recall the FoD process from Eq. (6):

$$\mathrm{d}x_t = \theta_t (\mu - x_t) \, \mathrm{d}t + \sigma_t (x_t - \mu) \, \mathrm{d}w_t. \tag{19}$$

To solve this SDE, we introduce a new variable $y_t = x_t - \mu$ and replace it into Eq. (19), which typically yields a Geometric Brownian motion (Ross, 2014) on $y_t$, given by

$$\mathrm{d}y_t = -\theta_t y_t \, \mathrm{d}t + \sigma_t y_t \, \mathrm{d}w_t. \tag{20}$$

To simplify the notation, we use $y$ rather than $y_t$ in all the following equations. This equation can be solved by applying Itô's formula:

$$\begin{aligned}
\mathrm{d}\psi(y, t) &= \frac{\partial \psi}{\partial t}(y, t) \, \mathrm{d}t + \frac{\partial \psi}{\partial y}(y, t) f(y, t) \, \mathrm{d}t \\
&\quad + \frac{1}{2} \frac{\partial^2 \psi}{\partial y^2}(y, t) g(t)^2 \, \mathrm{d}t \\
&\quad + \frac{\partial \psi}{\partial y}(y, t) g(t) \, \mathrm{d}w,
\end{aligned} \tag{21}$$

where $\psi(y, t) = \ln|y|$ is a surrogate differentiable function. By substituting $f(y, t)$ and $g(t)$ with the drift and the diffusion functions in (20), we obtain

$$\mathrm{d}\psi(y, t) = -(\theta_t + \frac{\sigma_t^2}{2}) \, \mathrm{d}t + \sigma_t \, \mathrm{d}w. \tag{22}$$

Then we can solve $y_t$ conditioned on $y_s$, by integrating both sides:

$$\ln|y_t| - \ln|y_s| = -\int_s^t (\theta_z + \frac{\sigma_z^2}{2}) \, \mathrm{d}z + \int_s^t \sigma_z \mathrm{d}w(z) \tag{23}$$

where the stochastic interaction follow a Gaussian distribution, i.e., $\int_s^t \sigma_z \, \mathrm{d}w(z) \sim \mathcal{N}\left(0, \int_s^t \sigma_z^2 \mathrm{d}z\right)$, then we can rewrite:

$$\ln|y_t| = \ln|y_s| - (\bar{\theta}_{s:t} + \frac{\bar{\sigma}_{s:t}^2}{2}) + \bar{\sigma}_{s:t} \epsilon_{s \to t}, \quad \epsilon_{s \to t} \sim \mathcal{N}(0, I), \tag{24}$$

where $\bar{\theta}_{s:t} = \int_s^t \theta_z \, \mathrm{d}z$, $\bar{\sigma}_{s:t}^2 = \int_s^t \sigma_z^2 \, \mathrm{d}z$, and $\bar{\sigma}_{s:t} = \sqrt{\bar{\sigma}_{s:t}^2}$. By replacing $y_t$ with the original $x_t - \mu$, we have the following

$$\ln|x_t - \mu| = \ln|x_s - \mu| - (\bar{\theta}_{s:t} + \frac{\bar{\sigma}_{s:t}^2}{2}) + \bar{\sigma}_{s:t} \epsilon_{s \to t}. \tag{25}$$

Note that the sign of $x_t - \mu$ remains consistent across all times $t$, therefore, we can safely omit the absolute value inside the logarithm, which leads to a log-normal distribution:

$$\log(\mu - x_t) \sim \mathcal{N}\left(\log(\mu - x_s) - \int_s^t \left(\theta_z + \frac{1}{2}\sigma_z^2\right) \mathrm{d}z, \ \int_s^t \sigma_z^2 \, \mathrm{d}z \, I\right)., \tag{26}$$

which gives the Corollary 3.2. In addition, applying the exponential function to both sides yields

$$(x_t - \mu) = (x_s - \mu) \mathrm{e}^{-(\bar{\theta}_{s:t} + \frac{\bar{\sigma}_{s:t}^2}{2}) + \bar{\sigma}_{s:t} \epsilon_{s \to t}} \tag{27}$$

which is the solution to the SDE, and thus we complete the proof. $\qquad\square$

## A.2 PROOF OF THE KL DIVERGENCE

The KL divergence of (10) can be derived as follows:

$$
\tilde{L} := -\log p_\phi(x_T)
$$

$$
= -\log \int p_\phi(x_{0:T}) \, \mathrm{d}x_{0:T-1}
$$

$$
= -\log \int \frac{p_\phi(x_{0:T}) q(x_{0:T-1} \mid x_T)}{q(x_{0:T-1} \mid x_T)} \, \mathrm{d}x_{0:T-1}
$$

$$
= -\log \mathbb{E}_{q(x_{0:T-1}|x_T)} \left[ \frac{p_\phi(x_{0:T})}{q(x_{0:T-1} \mid x_T)} \right]
$$

$$
\leq \underbrace{\mathbb{E}_{q(x_{0:T-1}|x_T)} \left[ -\log \frac{p_\phi(x_{0:T})}{q(x_{0:T-1} \mid x_T)} \right]}_{\textit{negative evidence lower bound (ELBO)}} \qquad \text{(Jensen's Inequality)}
$$

$$
= \mathbb{E}_{q(x_{0:T-1}|x_T)} \left[ -\log \frac{p(x_0) \prod_{t=1}^{T} p_\phi(x_t \mid x_{t-1})}{\prod_{t=1}^{T} q(x_{t-1} \mid x_t)} \right]
$$

$$
= \mathbb{E}_{q(x_{0:T-1}|x_T)} \left[ -\log p(x_0) - \sum_{t=1}^{T} \log \frac{p_\phi(x_t \mid x_{t-1})}{q(x_{t-1} \mid x_t)} \right]
$$

$$
= \mathbb{E}_{q(x_{0:T-1}|x_T)} \left[ -\log p(x_0) - \sum_{t=1}^{T-1} \log \underbrace{\frac{p_\phi(x_t \mid x_{t-1})}{q(x_t \mid x_{t-1}, x_T)} \cdot \frac{q(x_t \mid x_T)}{q(x_{t-1} \mid x_T)}}_{\text{Bayes' rule on } q(x_{t-1}|x_t, x_T)} - \log \frac{p_\phi(x_T \mid x_{T-1})}{q(x_{T-1} \mid x_T)} \right]
$$

$$
= \mathbb{E}_{q(x_{0:T-1}|x_T)} \left[ -\log p(x_0) - \sum_{t=1}^{T-1} \log \frac{p_\phi(x_t \mid x_{t-1})}{q(x_t \mid x_{t-1}, x_T)} - \log \frac{q(x_{T-1} \mid x_T)}{q(x_0 \mid x_T)} - \log \frac{p_\phi(x_T \mid x_{T-1})}{q(x_{T-1} \mid x_T)} \right]
$$

$$
= \mathbb{E}_{q(x_{0:T-1}|x_T)} \left[ -\log \frac{p(x_0)}{q(x_0 \mid x_T)} - \sum_{t=1}^{T-1} \log \frac{p_\phi(x_t \mid x_{t-1})}{q(x_t \mid x_{t-1}, x_T)} - \log p_\phi(x_T \mid x_{T-1}) \right]
$$

$$
= D_{KL}(q(x_0 \mid x_T) \,\|\, p(x_0)) \qquad (D_{KL}(p\|q) = \mathbb{E}[-\log \tfrac{p}{q}])
$$

$$
+ \sum_{t=1}^{T-1} D_{KL}(q(x_t \mid x_{t-1}, x_T) \,\|\, p_\phi(x_t \mid x_{t-1})) - \mathbb{E}_{q(x_{T-1}|x_T)} \left[ \log p_\phi(x_T \mid x_{T-1}) \right],
$$

$$
\tag{28}
$$

where the first term can be ignored since it doesn't have trainable parameters, and the third term can be merged to the final stochastic flow matching objective.

## A.3 DERIVATION OF STOCHASTIC FLOW MATCHING

**Remark.** For two log-normal distributions with $\log p_1 \sim \mathcal{N}(\mu_1, \sigma_1^2)$ and $\log p_2 \sim \mathcal{N}(\mu_2, \sigma_2^2)$, the KL divergence between them is given by

$$
D_{\mathrm{KL}}(p_1 \| p_2) = \frac{(\mu_1 - \mu_2)^2}{2\sigma_2^2} + \frac{\sigma_1^2}{2\sigma_2^2} + \ln \frac{\sigma_2}{\sigma_1} - \frac{1}{2}. \tag{29}
$$

This result is for scalars but can be naturally extended to high-dimensional cases. In the following, we will use it to derive our stochastic flow matching objective.

More specifically, given the KL divergence

$$
\mathbb{E}_p \left[ \sum_{t=0}^{T-1} D_{KL}(p(f_\mu(x_{t+1}) \mid f_\mu(x_t) \,\|\, p(f_\phi(x_{t+1}, t) \mid f_\phi(x_t, t)) \right] \tag{30}
$$

and the truth that $f_\phi(x_t, t)$ approximates $\mu - x_t$. Since the two transitions are log-normally distributed and share the same parameters $\theta_t$ and $\sigma_t$ as in Eq. (8), we can initially obtain a log space loss based

on the Remark (29), as

$$L := \mathbb{E}_{\mu \sim p_{\text{data}}, x_t \sim q(x_t|x_0,\mu)} \left[ \frac{1}{2\sigma_{t+1}^2} \| \log f_\mu(x_t, t) - \log f_\phi(x_t, t) \|^2 \right]. \tag{31}$$

However, this would run into issues when $\mu - x_t \leq 0$. Though this can be alleviated using absolute values and adding a small additive term $\epsilon$, it complicates the learning and is still unstable.

To address it, we assume that, after some training, $f_\phi(x_t, t)/f_\mu(x_t, t) = 1 + \delta$ where $|\delta| \ll 1$. The first-order Taylor approximation is then

$$\log f_\phi(x_t, t) - \log f_\mu(x_t, t) = \log(1 + \delta) \approx \frac{f_\phi(x_t, t) - f_\mu(x_t, t)}{f_\mu(x_t, t)}. \tag{32}$$

Since the denominator does not depend on the parameters, we obtain our simplified loss function by omitting all non-trainable weights:

$$L := \mathbb{E}_{\mu \sim p_{\text{data}}, x_t \sim q(x_t|x_0,\mu)} \left[ \| f_\mu(x_t, t) - f_\phi(x_t, t) \|^2 \right], \tag{33}$$

which is the proposed stochastic flow matching objective.

Note that the first-order Taylor approximation is not valid during the early stages of training, when the predicted flow is typically far from the ground truth. In such cases, Eq. (12) no longer corresponds to an exact KL objective, but instead serves as a surrogate loss for directly learning the flow. Nevertheless, we emphasize that this surrogate objective can empirically achieve strong performance and simplify the optimization. This is conceptually analogous to denoising objectives and those used in score matching (not exact KL objectives, but have still proven effective in practice). Table 5 below tracks both the approximation error $(\log(1 + \delta) - \delta)$ and magnitude of the higher-order terms $(-\frac{1}{2}\delta^2)$. As one can see, the approximation is loose during the early stages of training but becomes valid $(\approx 0.1)$ after $\sim 50,000$ iterations.

Table 5: Tracking the approximation error and magnitude of the higher-order terms.

| Training steps | Approximation error | Magnitude |
|---|---|---|
| 1,000 | 0.404 | 0.333 |
| 10,000 | 0.183 | 0.224 |
| 50,000 | 0.105 | 0.108 |
| 100,000 | 0.095 | 0.086 |

## B  MAXIMUM LIKELIHOOD ESTIMATION

Following DDPMs (Ho et al., 2020), both our training and sampling are implemented with discrete times, which can of course be converted into continuous times, as in Score SDEs (Song et al., 2021), but that requires $\theta$ and $\sigma$ schedules to be integrable. Below, we further show that the solution to FoD also allows us to compute the maximum likelihood:

Given the clean data $\mu$, assuming that there exists an optimal forward transition from $z_t$ to $z_{t+1}$, where $z_t = |\mu - x_t|$. In other words, we want to maximize the likelihood of $p(z_{t+1} \mid z_t)$, which is a log-normal distribution as illustrated in Corollary 3.2, and its density is given by

$$p(z_{t+1} \mid z_t) = \frac{1}{z_{t+1}\sigma_{t+1}\sqrt{2\pi}} \exp\left( -\frac{1}{2\sigma_{t+1}^2} \left[ \ln z_{t+1} - lnz_t + \left(\theta_{t+1} + \frac{\sigma_{t+1}^2}{2}\right) \right]^2 \right). \tag{34}$$

Then, we can minimize the negative log-likelihood:

$$-\ln p(z_{t+1} \mid z_t) = \ln z_{t+1} + \frac{1}{2\sigma_{t+1}^2} \left[ \ln z_{t+1} - lnz_t + \left(\theta_{t+1} + \frac{\sigma_{t+1}^2}{2}\right) \right]^2 + \ln(\sigma_{t+1}\sqrt{2\pi}), \tag{35}$$

which can be solved by setting the gradient to 0, as

$$\nabla_{z_{t+1}} - \ln p(z_{t+1} \mid z_t) = \frac{1}{z_{t+1}} \left[ 1 + \frac{1}{\sigma_{t+1}^2} \left( \ln z_{t+1} - lnz_t + \left(\theta_{t+1} + \frac{\sigma_{t+1}^2}{2}\right) \right) \right] = 0. \tag{36}$$

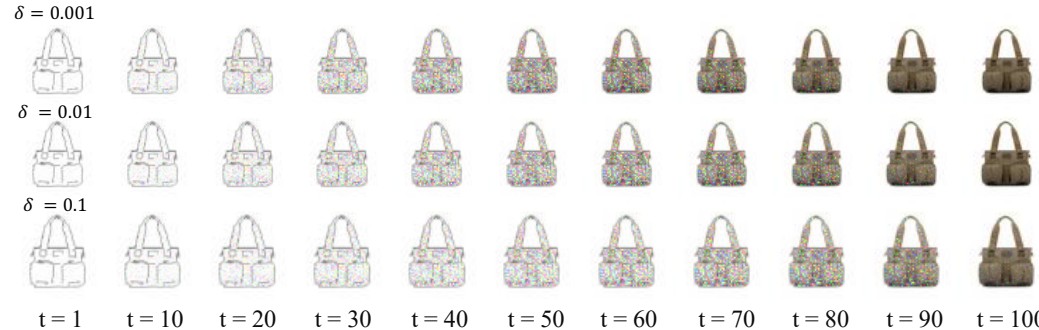

$\delta = 0.001$

$\delta = 0.01$

$\delta = 0.1$

$t = 1 \quad t = 10 \quad t = 20 \quad t = 30 \quad t = 40 \quad t = 50 \quad t = 60 \quad t = 70 \quad t = 80 \quad t = 90 \quad t = 100$

Figure 7: Comparison of the sampling process of FoD with different time interval coefficient values.

Since $z_{t+1}$ is not zero, the optimal solution $z^*_{t+1}$ is obtained according to

$$z_{t+1} = z_t e^{-(\theta_{t+1} + \frac{\sigma^2_{t+1}}{2}) - \sigma^2_{t+1}}. \tag{37}$$

Recall that $\mu - x_t$ has the same sign for all times $t$. Replacing $z_t$ with $|\mu - x_t|$ gives the following:

$$(\mu - x_{t+1})^* = (\mu - x_t)e^{-(\theta_{t+1} + \frac{\sigma^2_{t+1}}{2}) - \sigma^2_{t+1}}, \tag{38}$$

which is the optimal forward flow from $x_{t+1}$ to $\mu$.

Based on it, we can get the maximum likelihood learning objective:

$$L = \left\| (\mu - x_{t+1})^* - \mathbb{E}[\mu_\phi - x_{t+1}] \right\|^2, \tag{39}$$

$$L = \left\| x^*_{t+1} - \mathbb{E}_\phi[x_{t+1} \mid x_t] \right\|^2, \tag{40}$$

where $\mathbb{E}_\phi[x_{t+1} \mid x_t]$ is the expectation given $x_t$ in discrete time: $\mathbb{E}_\phi[x_{t+1} \mid x_t] = x_t + \mathrm{d}x_t$.

$$\begin{aligned} \mathbb{E}[\mu_\phi - x_{t+1}] &= (\mu_\phi - x_t)e^{-(\theta_{t+1} + \frac{\sigma^2_{t+1}}{2}) + \frac{\sigma^2_{t+1}}{2}} \\ &= (\mu_\phi - x_t)e^{-\theta_{t+1}}. \end{aligned} \tag{41}$$

Combining (38) and (41) predicts $x_{t+1}$ in the optimal path. This maximum likelihood-based loss function performs similarly to stochastic flow matching and can be potentially used in future work.

## C    MORE EXPERIMENTAL DETAILS

### C.1    IMPLEMENTATION AND DATASETS

We set the number of diffusion steps to 100 for all tasks. Practically, we choose to use a discrete time implementation for our method, where we let $\bar{\theta}_t = \int_0^t \theta_z \, \mathrm{d}z \approx \sum \theta_t \Delta t$ and $\bar{\sigma}^2_t = \int_s^t \sigma^2_z \, \mathrm{d}z \approx \sum \sigma^2_t \Delta t$. To ensure that FoD converges to the clean data $\mu$, we let the deterministic exponential term at the terminal state be a smaller value, i.e. $e^{-\int_0^t (\theta_s + \frac{1}{2}\sigma^2_s) \, \mathrm{d}s} = \delta = 0.001$. Solving it leads to an updated time interval $\Delta t = \frac{\log \delta}{\int_0^t (\theta_s + \frac{1}{2}\sigma^2_s) \, \mathrm{d}s}$[2]. Note that the small value of the exponential term guarantees the terminal state converges to the clean target data. This also indicates that large values are required for the FoD's drift and diffusion coefficients, i.e. $\theta_t$ and $\sigma_t$, ensuring the stochasticity and that sufficient noise is added to the sampling process. We visualize the sampling process with different $\delta$ values in Figure 7. As can be observed, the diffusion volatility remains similar across different $\delta$ values, while processes with smaller coefficients converge more rapidly toward the clean target data. Overall, the sampling trajectories appear temporally shifted (or horizontally scaled) as $\delta$ increases from 0.001 to 0.1. For image-to-image translation, most datasets are the same as in Pix2Pix (Isola et al., 2017), but with all images resized to $64 \times 64$ to improve training and testing

---

[2]Our code is provided in the supplementary material.

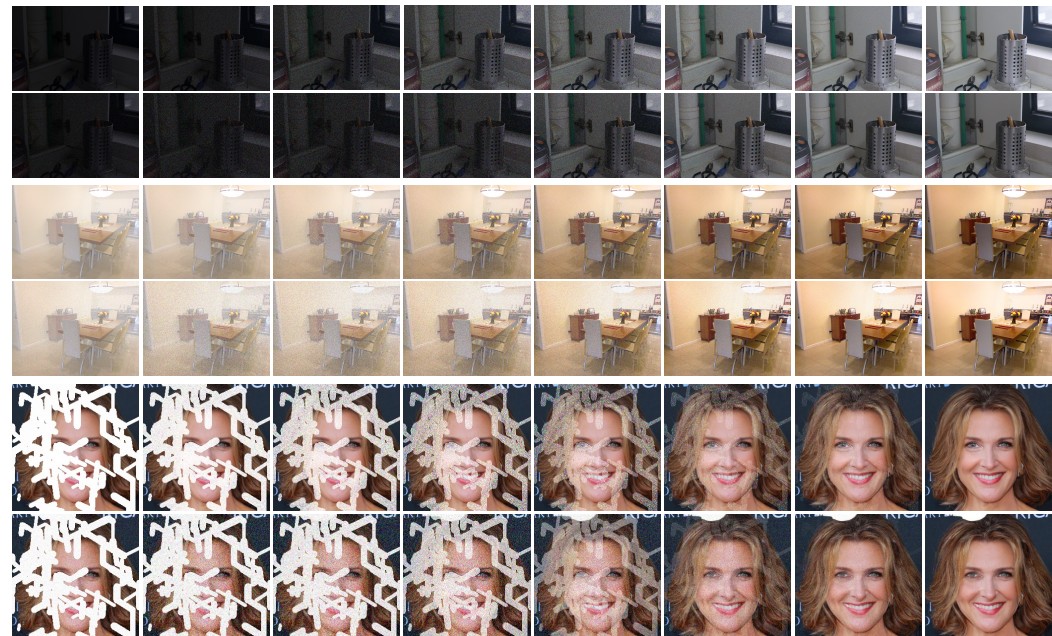

Figure 8: Comparison of the sampling process on different image restoration tasks. In each task, the top row is FoD and the bottom row is GOUB (Yue et al., 2024). Both methods gradually inject noise and subsequently denoise these intermediate states. The diffusion bridge adds noise across the entire image for realistic generation. In contrast, our FoD focuses and adds noise to the degradation areas.

efficiency. All these image-to-image translation experiments share the same settings, as the goal is to illustrate the general applicability of FoD rather than to optimize performance for each task.

In addition, the details of image restoration datasets are listed below:

- *Deraining*: collected from the Rain100H (Yang et al., 2017) dataset containing 1800 images for training and 100 images for testing.
- *Dehazing*: collected from the RESIDE-6k (Qin et al., 2020) dataset which has mixed indoor and outdoor images with 6000 images for training and 1000 images for testing.
- *Low-light enhancement*: collected from the LOL (Wei et al., 2018) dataset containing 485 images for training and 15 images for testing.
- *Face inpainting*: we use CelebaHQ as the training dataset and divide 100 images with 100 thin masks from RePaint (Lugmayr et al., 2022) for testing.

## C.2 ADDITIONAL DISCUSSIONS

**Fast Sampling**   As mentioned in Section 3.3, we provide two fast sampling strategies with Markov and non-Markov Chains. Their comparison on image deraining is illustrated in Section 5 of the main paper. Here, we give more comparison results on four image restoration tasks, reporting PSNR, SSIM, LPIPS, and FID values in Figure 13. The results with standard FoD sampling (See Algorithm 2) are also reported as the baseline. It can be observed that, in most tasks, decreasing the number of steps in fast sampling approaches leads to better performance, particularly in terms of PSNR and SSIM. Their perceptual performance also decreases when the number of steps decreases from 20 to 5, which suggests we set the sample step to 10 as a practical rule of thumb for efficient sampling. Note that the key difference between faster sampling approaches and the standard FoD sampling is that the latter's iterative process highly depends on the $\mu$ prediction, while the former gradually refines the state $x_t$ rather than $\mu$. In practice, the MC sampler is preferable for highly ill-posed problems (e.g., image-to-image translation) while the non-MC sampler is more suitable for structure-preserving tasks such as image restoration. The visual comparisons are illustrated in Figure 14.

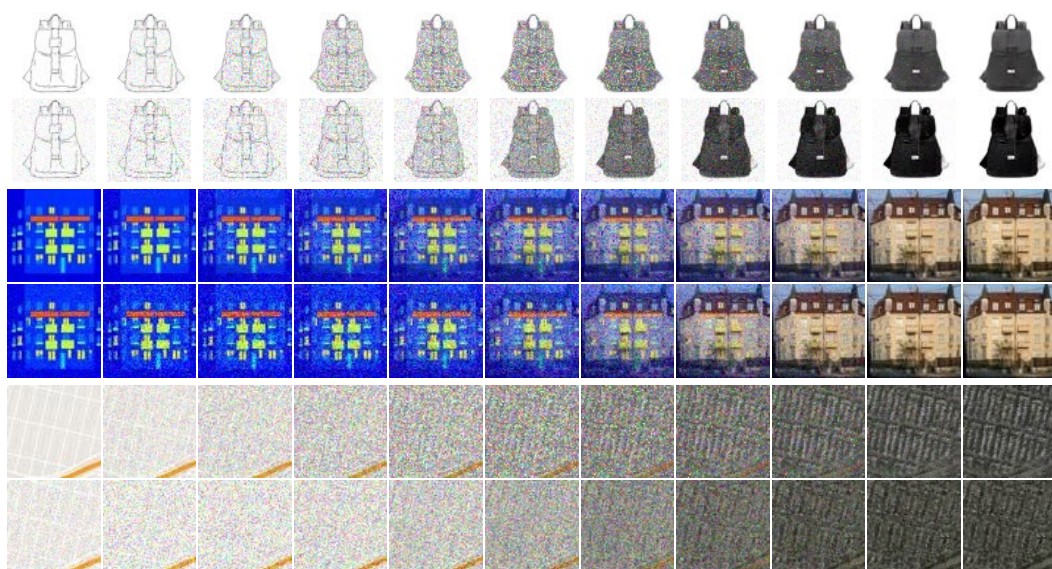

Figure 9: Comparison of the sampling process on different image-to-image translation tasks. In each task, the top row is FoD and the bottom row is GOUB (Yue et al., 2024). Both methods gradually inject noise and subsequently denoise these intermediate states. While the diffusion bridge model GOUB adds noise across the entire image in all cases, our FoD focuses only on areas that actually need to be transformed, as for the *edges to handbags* translation example in the top row.

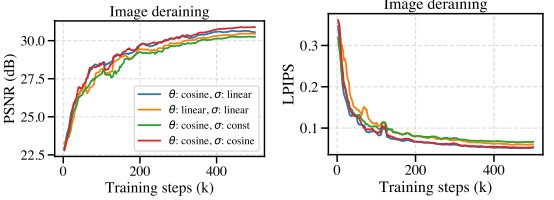

Figure 10: Training curves of different $\theta$ and $\sigma$ schedules.

Table 6: Analysis of different choices of $\theta$ and $\sigma$ schedules on image deraining.

| Method | PSNR↑ | SSIM↑ | LPIPS↓ | FID↓ |
|---|---|---|---|---|
| IR-SDE (baseline) | 31.65 | 0.904 | 0.047 | 18.64 |
| linear $\theta$, linear $\sigma$ | 32.48 | 0.918 | 0.045 | 16.11 |
| cosine $\theta$, const $\sigma$ | 32.22 | 0.906 | 0.046 | 17.34 |
| cosine $\theta$, linear $\sigma$ | 32.56 | 0.925 | 0.038 | 14.10 |
| cosine $\theta$, cosine $\sigma$ | 32.73 | 0.931 | 0.039 | 15.12 |

**Illustration of the Diffusion Process**    To clearly show the noise injection process with our model, we apply the trained FoD on various tasks, including both image restoration and unconditional image generation, and illustrate their intermediate states along timesteps as shown in Figure 8 and Figure 9. It can be observed that, for image restoration tasks, the noise level for each task and even for each image is different. This is given by the term $\mu - x_0$ in the solution, where areas with large difference (between LQ and HQ images) tend to produce large noise. It also means our model focuses more on restoring the degradations instead of reconstructing the whole image, yielding a more efficient solution to the image restoration problem. We also find that the noise is only injected into the degraded areas, such as the masked regions in deraining and face inpainting. Figure 11 provides additional training curves of our FoD and its noise-free variant on different tasks to illustrate the significance of noise injection in image restoration. More examples for the FoD diffusion process are provided in Figure 15.

**Choice of $\theta$ and $\sigma$ schedules**    We employ the commonly used cosine and linear schedules for $\theta_t$ and $\sigma_t$, respectively, for all previous experiments. Here, we provide additional training curves, in terms of PSNR and LPIPS, and comprehensive evaluations of different schedule combinations on the deraining task, as shown Figure 10 and Table 6, to analyze the effect of the choice of $\theta, \sigma$ schedules. All method variants outperform the diffusion baseline model IR-SDE (Luo et al., 2023a), demonstrating the robustness and stability of our FoD framework to various combinations of schedules. In addition, it is observed that using a constant $\sigma$ achieves slightly worse results on all metrics, while applying cosine schedules to both parameters yields the best performance.

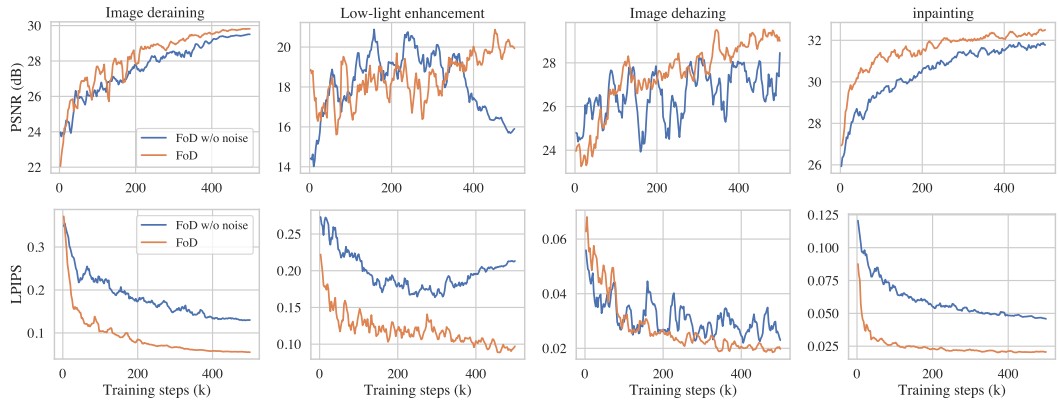

Figure 11: Training curves of our FoD and its noise-free variant on different tasks in terms of PSNR and LPIPS. All these results demonstrate the effectiveness of noise injection in image restoration.

| Method | FID↓ |
|---|---|
| DDPM | 3.17 |
| DDPM* | 4.36 |
| Score SDE | 2.38 |
| NCSN | 25.32 |
| NCSNv2 | 10.87 |
| Flow Matching w/ diff path | 10.31 |
| Flow Matching w/ OT path | 6.96 |
| Rectified Flow | 2.58 |
| FoD-SDE (T=100) | 7.89 |
| FoD-SDE (T=1000) | 6.59 |
| FoD-ODE (T=100) | 5.01 |
| FoD-ODE (T=1000) | 4.60 |
| FoD-ODE w/o $\alpha$ (T=1000) | 4.33 |

FoD-SDE      FoD-ODE

Figure 12: **Left:** Visual results of unconditional generation on the CIFAR-10 dataset by two variants of our FoD model. **Right:** Quantitative comparison of our methods with other approaches on CIFAR-10. Here, '*' means a re-implementation using our U-Net architecture and hyperparameter setting.

### C.3 Unconditional Image Generation

In this section, we evaluate the unconditional generation performance of our model on the CIFAR-10 dataset (Krizhevsky et al., 2009), using the same architecture as in image-conditioned generation (i.e., attention is not used). Specifically, we showcase results from both our FoD model and its ODE-based variant, FoD-ODE (see Section 3.4). Both models share the same $\theta$ schedule and are sampled with 100 steps. We compare against several baselines, including 1) diffusion models with forward-backward schemes such as DDPM (Ho et al., 2020) and Score SDE (Song et al., 2021), as well as 2) forward-only frameworks: score-based generative models (SGMs) like NSCN (Song & Ermon, 2019) and NSCNv2 (Song & Ermon, 2020), flow matching generative models using diffusion and optimal transport (OT) paths (Lipman et al., 2022), and Rectified Flow (Liu et al., 2022) which also adopts the OT path for data transformation.

**Results** We present the generated image samples and quantitative comparisons in Figure 12, with visual examples on the left and numerical results summarized in the table on the right. Under the forward-only framework, our FoD model with SDE sampling achieves a competitive FID of 7.89, outperforming other score-based generative

| Method | FID↓ |
|---|---|
| FoD-SDE + Log-Normal prior | 16.45 |
| FoD-SDE + Normal prior | 7.89 |

Table 7: FoD-SDE with different priors for unconditional image generation.

models as well as the flow matching approach using diffusion paths. The ODE-based variant of FoD improves the performance to a FID of 5.01, surpassing the flow matching model based on the optimal transport path. Similar to standard diffusion models, increasing the sampling steps to

1000 improves the performance. Moreover, adopting the rectified flow objective (i.e., ignoring $\alpha_t$ in (16)) further decreases the FID to 4.33. Nonetheless, our overall performance on CIFAR-10 remains inferior to Rectified Flow and conventional diffusion models using a forward-backward scheme. We partly attribute this performance gap to differences in architectural choices, hyperparameter tuning, etc. Considering that FoD adds multiplicative noise that follows a log-normal distribution at each step (Proposition 3.1), we include an additional experiment to evaluate whether using a log-normal prior could improve unconditional generation. As shown in Table 7, replacing the standard Normal prior with Log-Normal prior actually degrades the performance. We attribute this to the fact that the log-normal distribution enforces samples to be strictly positive, which is incompatible with FoD's state space where $x_t$ can be both positive and negative, thereby leading to a mismatch between the prior and the FoD forward process. This experiment suggests that a standard normal prior remains more suitable for FoD despite its multiplicative SDE structure. We still believe that there are other potential directions to explore to improve the unconditional generation capabilities, such as log-space transformations and optimal transport-based drift paths, which we plan to investigate in future work. However, we also note that the main aim of this paper is to construct an effective single diffusion process model, which we expect to be particularly well-suited for image-conditioned generation tasks such as image restoration.

Table 8: Comparison of different approaches (FM (Lipman et al., 2022), GOUB (Yue et al., 2024), UniDB (Zhu et al., 2025), and our FoD) on four image-to-image translation tasks. Here, 'FM*' means it is implemented using the same architecture and parameter settings as our FoD.

| Method | Edges to handbags | | | Labels to Facade | | | Map to Photo | | | Night to Day | | |
|---|---|---|---|---|---|---|---|---|---|---|---|---|
| | MSE↓ | LPIPS↓ | FID↓ | MSE↓ | LPIPS↓ | FID↓ | MSE↓ | LPIPS↓ | FID↓ | MSE↓ | LPIPS↓ | FID↓ |
| FM* | 0.105 | 0.248 | 25.29 | 0.0021 | 0.015 | 21.30 | 0.0049 | 0.0273 | 36.07 | 0.205 | 0.296 | 78.52 |
| GOUB | 0.054 | 0.224 | 9.36 | 0.0004 | 0.0014 | 3.47 | 0.0009 | 0.0069 | 11.62 | 0.162 | 0.241 | 64.35 |
| UniDB | 0.048 | 0.218 | 9.12 | 0.0003 | 0.0009 | 2.79 | 0.0008 | 0.0068 | 11.78 | 0.159 | 0.241 | 64.62 |
| FoD | 0.025 | 0.198 | 8.45 | 0.00004 | 0.0002 | 0.86 | 0.0001 | 0.0017 | 2.33 | 0.111 | 0.205 | 52.11 |

## C.4 ADDITIONAL RESULTS

In this section, we provide more results for four image restoration tasks including image deraining, low-light enhancement, image dehazing, and image inpainting in Figure 16, Figure 17, Figure 18, and Figure 19. In most tasks, the results produced by our method are sharper and more realistic. For image-to-image translation, we report the comprehensive evaluations (including MSE, LPIPS, and FID metrics) for our FoD and other flow matching (Lipman et al., 2022) and diffusion bridge (GOUB (Yue et al., 2024) and UniDB (Zhu et al., 2025)) models on the 1) edges to handbags, 2) labels to facades, 3) maps to photos, and 4) night to day training datasets in Table 8. Here, we use sub-datasets of *edges to handbags* and *night to day* since their original training datasets are too large. In addition, the flow matching approach (FM) is implemented using the same architecture and parameter settings as our FoD. The results further prove the importance of noise injection in image-conditioned generation. In addition, Table 9 illustrates the comparison of our FoD with other approaches (SDEdit (Meng et al., 2022), Rectified Flow (Liu et al., 2022), and GOUB (Yue et al., 2024)) on the entire *edges to handbags* training dataset. For unconditional generation, more results of our model with SDE and ODE on CIFAR-10 are provided in Figure 20 and Figure 21, respectively. Please zoom in for the best view.

Table 9: Comparison of our method with other diffusion (SDEdit (Meng et al., 2022)), flow matching (Rectified Flow (Liu et al., 2022)), and diffusion bridge (GOUB (Yue et al., 2024) and UniDB (Zhu et al., 2025)) models on the *edges to handbags* training dataset.

| Method | MSE↓ | LPIPS↓ | FID↓ |
|---|---|---|---|
| SDEdit (Meng et al., 2022) | 0.510 | 0.271 | 26.5 |
| Rectified Flow (Liu et al., 2022) | 0.088 | 0.241 | 25.3 |
| GOUB (Yue et al., 2024) | 0.054 | 0.224 | 9.36 |
| UniDB (Zhu et al., 2025) | 0.048 | 0.218 | 9.12 |
| FoD (Ours) | 0.025 | 0.198 | 8.45 |

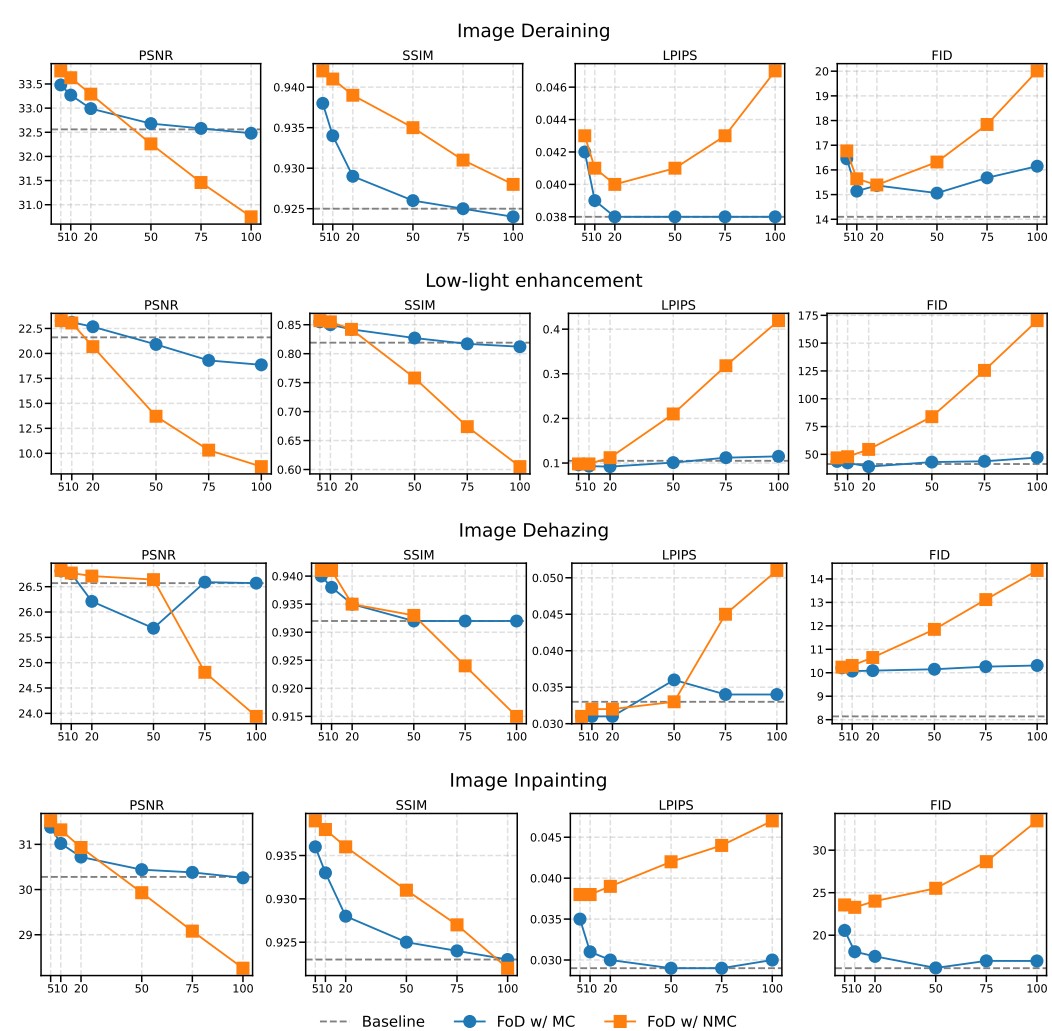

Figure 13: Comparison of different sampling approaches with pretrained FoD models on four image restoration tasks. The baseline is the Euler-Maruyama method with 100 sampling steps.

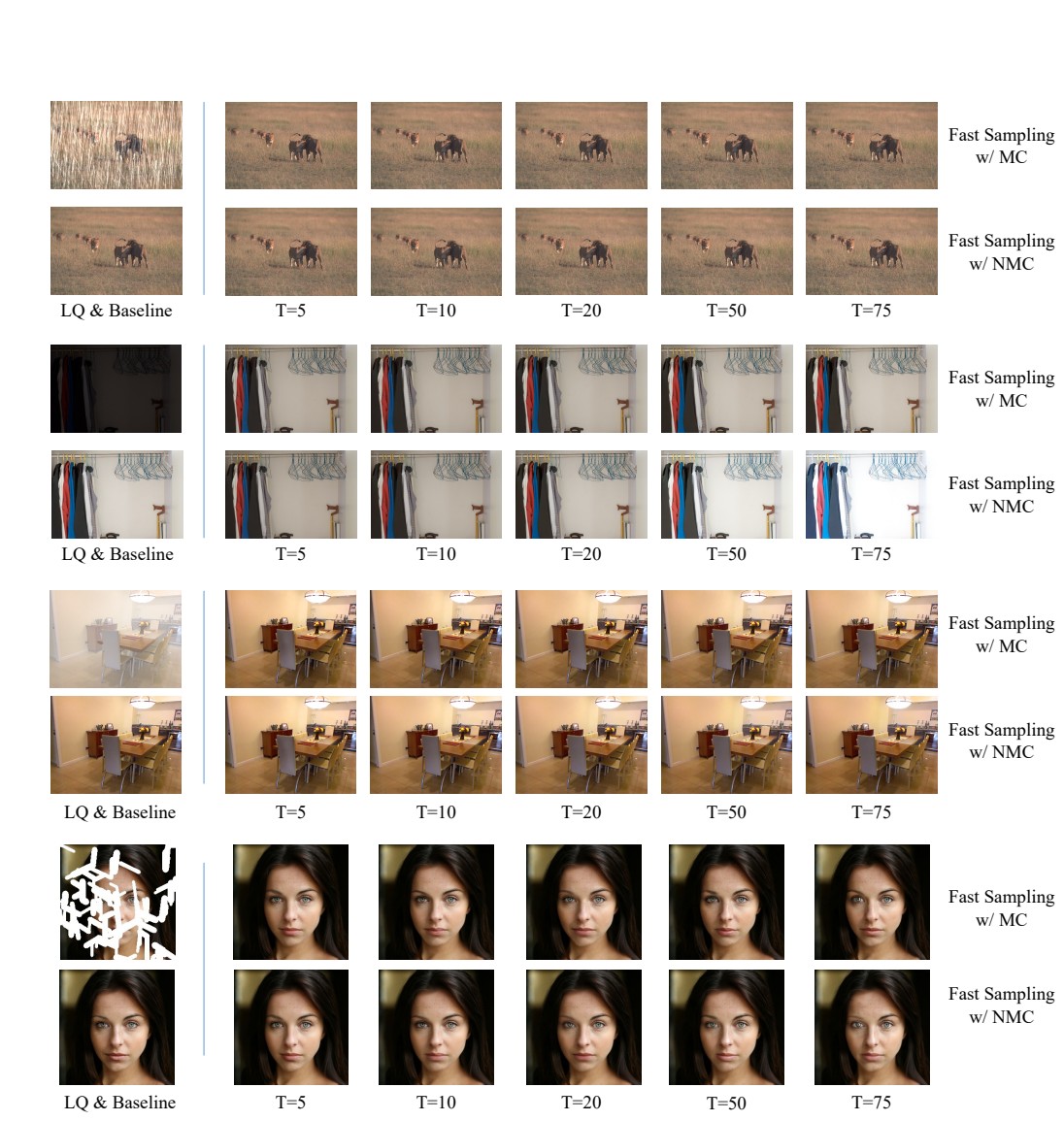

Figure 14: Comparison of fast sampling with Markov and non-Markov chains with different steps.

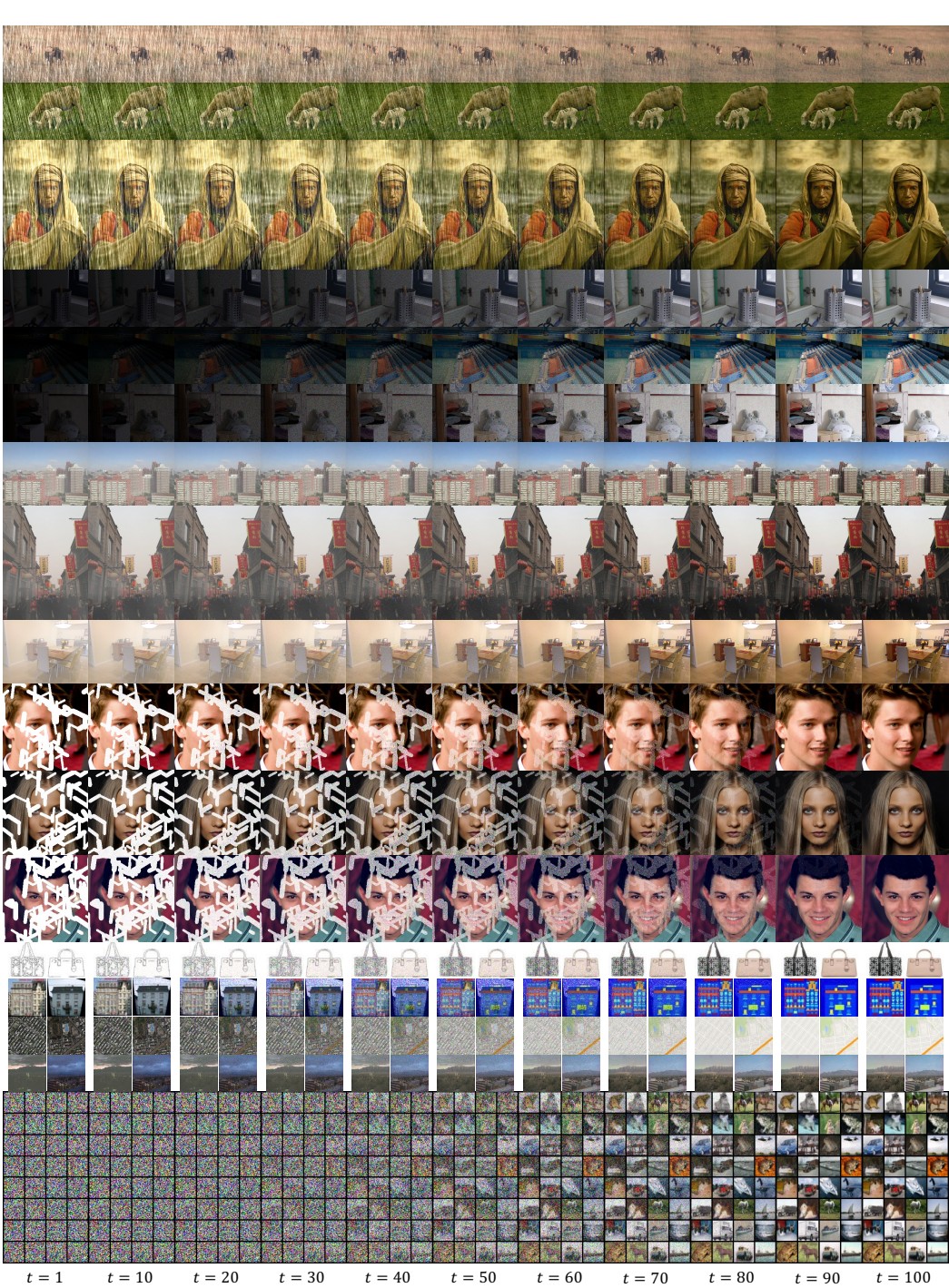

$t = 1$    $t = 10$    $t = 20$    $t = 30$    $t = 40$    $t = 50$    $t = 60$    $t = 70$    $t = 80$    $t = 90$    $t = 100$

Figure 15: Visualization of the diffusion process using trained FoD models on various tasks, including deraining, dehazing, low-light enhancement, face inpainting, and unconditional generation. In each case, FoD gradually injects noise into the degraded regions and subsequently denoises these intermediate states, restoring images with enhanced and corrected details.

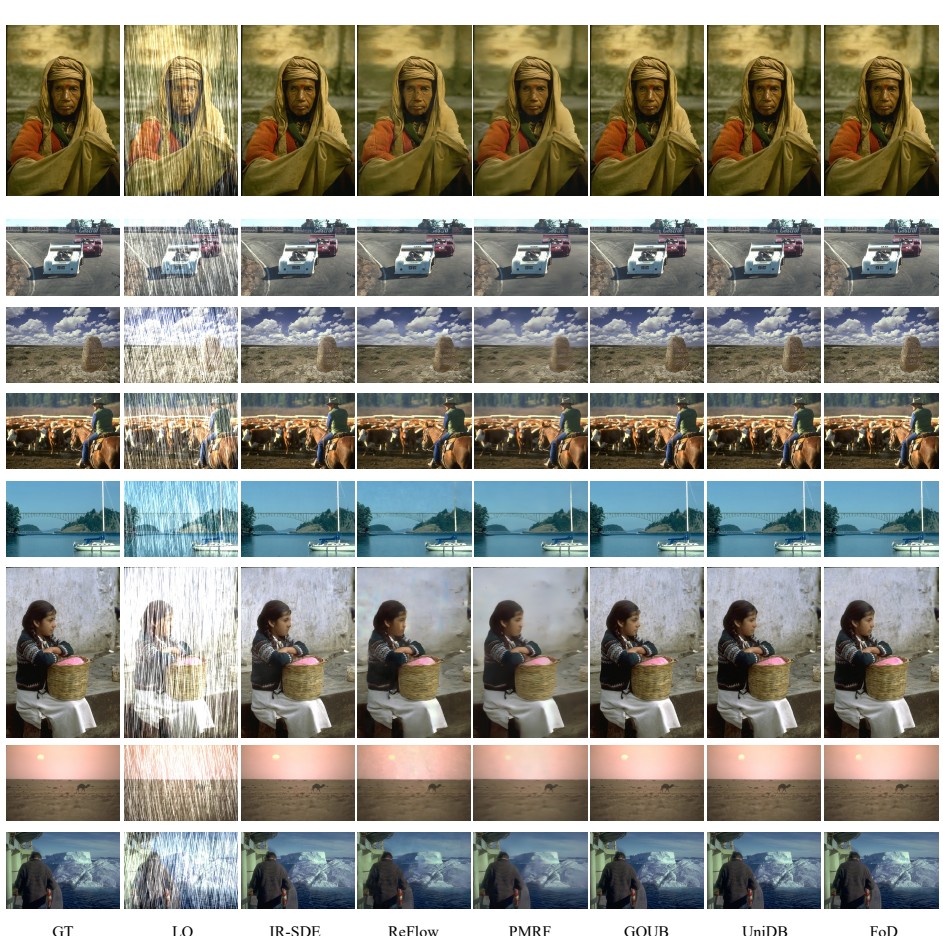

GT          LQ          IR-SDE          ReFlow          PMRF          GOUB          UniDB          FoD

Figure 16: Visual results of image deraining on Rain100H (Yang et al., 2017) dataset.

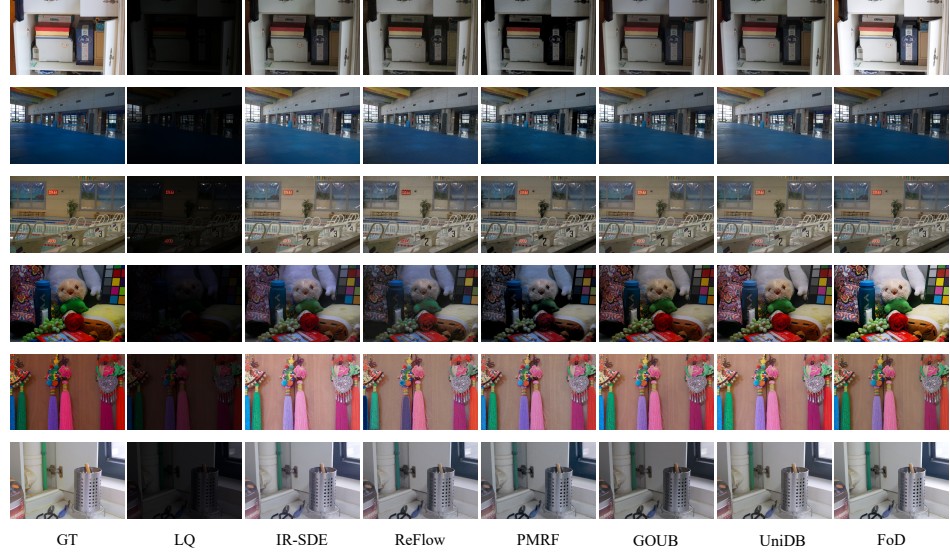

GT          LQ          IR-SDE          ReFlow          PMRF          GOUB          UniDB          FoD

Figure 17: Visual results of image low-light enhancement on LOL (Wei et al., 2018) dataset.

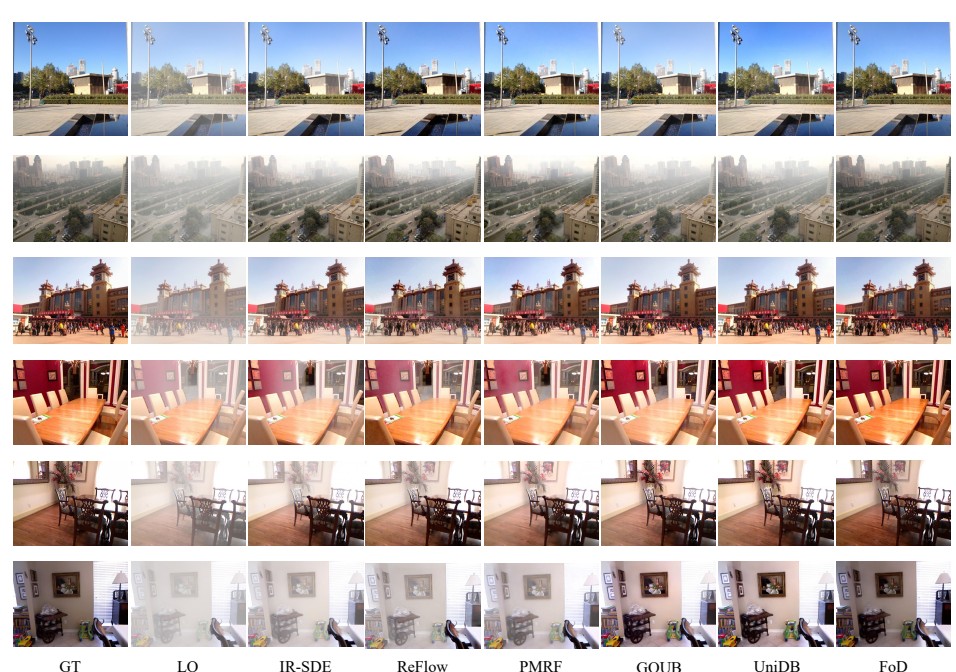

GT          LQ          IR-SDE      ReFlow      PMRF        GOUB        UniDB       FoD

Figure 18: Visual results of image dehazing on RESIDE-6k (Qin et al., 2020) dataset.

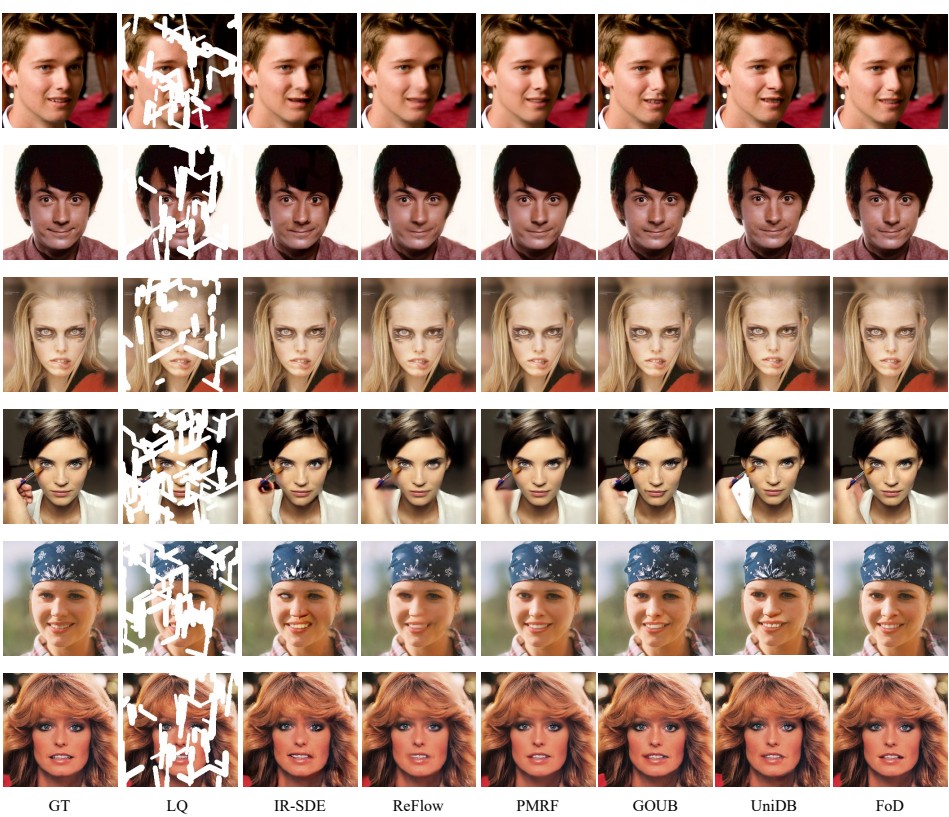

GT          LQ          IR-SDE      ReFlow      PMRF        GOUB        UniDB       FoD

Figure 19: Visual results of image inpainting on CelebA-HQ (Karras et al., 2017) dataset.

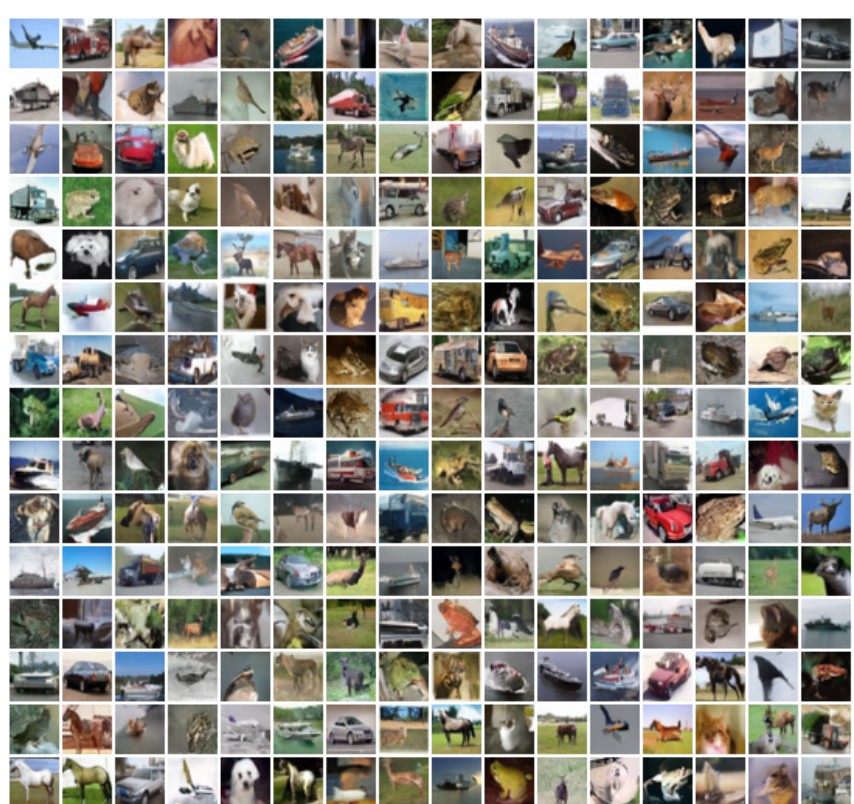

Figure 20: Unconditional generation by FoD (SDE sampler).

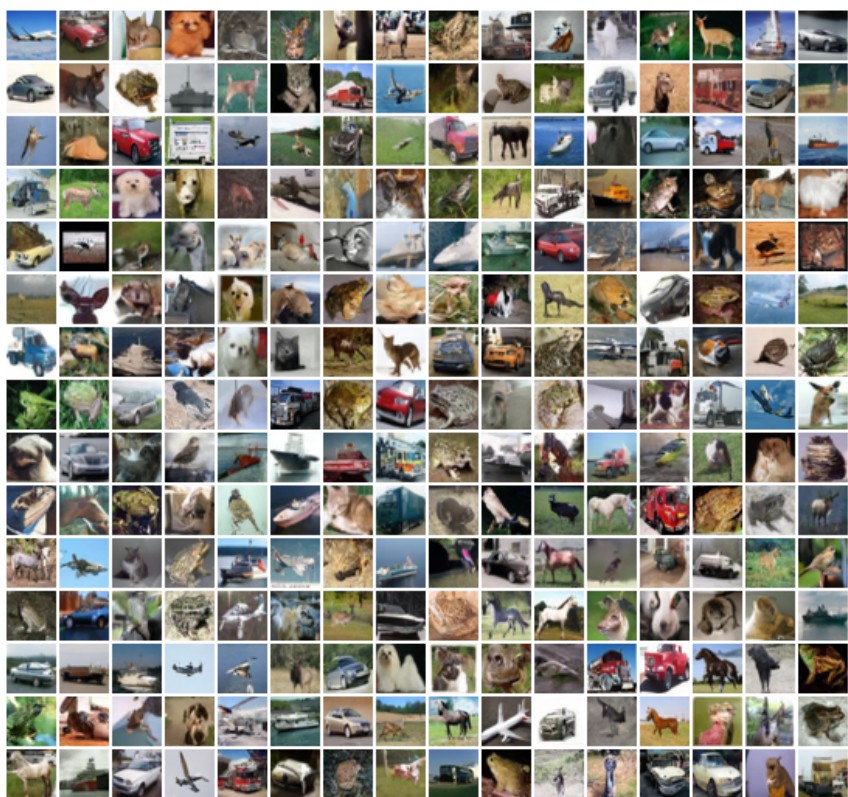

Figure 21: Unconditional generation by the ODE variant of FoD.

