# OpenReview forum: "Forward-only Diffusion Probabilistic Models"
_ICLR.cc/2026/Conference — Submitted to ICLR 2026_

### Official Review · Reviewer_nRYJ · 2025-10-28

**Soundness:** 2
**Presentation:** 3
**Contribution:** 1
**Rating:** 2
**Confidence:** 4

**Summary:**

This paper proposes a probabilistic forward-only diffusion model (FoD) that incorporates Geometric Brownian Motion (GBM) to transform an image into another that is in two distributions separately. The authors claimed that by introducing a mean-reversion term $\mu - x_t$​ into both the drift and diffusion coefficients of the SDE, they defined an analytically solvable forward-only process, eliminating the need to approximate or learn a reverse SDE. However, I have several concerned. First, I believe this claim of FoD is actually a backward diffusion (as I explained in Weakness section).  Second, in conventional diffusion models, the process involves diffusing an image  to noise (forward), and then reconstructing from noise to an image that closely remember the original image (reverse) although one can always  transform between two deterministic images. The setup in this paper actually resembles the GOUB., more recently; UNIDB. Although the authors proposed to  add the state dependency into the diffusion coefficient, there is no proof of the effect of this diffusion. Indeed the author almost has not discussion on the coefficients. Instead, the authors used extremely small diffusion coefficients, this made it actually more close to deterministic method, reducing the diffusion effects, which I believe making this paper relatively weak contribution to the field.

**Strengths:**

The paper is well written. The authors effort to consider diffusion coefficients as state dependent although the implementation is weak.

**Weaknesses:**

1. It seems the authors misinterpretation of forward diffusion as backward. This can be observed from they actually started from $x_T$ ∼ $p_{data}$ to  $x_0$ ∼$ p_{prior}$. The training step stated in the paper actually resembles the backward diffusion diffusion step in the conventional diffusion. There is not much different between this method in terms forward/backward as the diffusion itself is not a reason to stay on forward only as what we can see from GOUB.

2. P4. Line 178, as the author mentioned, "The subtractive form of the logarithm reflects that
the flow field decays multiplicatively from its initial value with a stochastic exponential scaling", this translate to the original image difference, the variance of the noise can explore, I wonder how the authors address this issue. The only solution in the paper is that they set the $ e^{-\int_0^t (\theta_s+\frac{1}{2} \sigma_s^2) ds}= 0.001$. The authors should have discussed the impact of selecting such small diffusion coefficients. This small value seems to make it almost a deterministic process.

3. As I mentioned in the summary and item 2, the authors used extremely small diffusion coefficients, this made it actually more close to deterministic method, reducing the diffusion effects. This can be observed in several sequences of images. For example, the second row images (over various time steps) in Figure 1 seemed so deterministic.

4. The following paper should be referred:
1). K Zhu et. al. UniDB: A Unified Diffusion Bridge Framework via Stochastic Optimal Control
2). G Kim et. al. Diffusion-based generative model for financial time series via geometric Brownian motion

5.  PROPOSITION 3.1 seems like an existing result and no need to prove again.
6. Although the paper compared their method to GOUB in terms of metrics such as FID, they did not provide the GOUB results in the figures of images that compares the results. It will be good to seem their comparisons in images as they are similar diffusions between two deterministic images.

**Questions:**

1. The authors need to discuss more how selection of the coefficients impact the behavior of the diffusion.
2. Choice of $\theta$ and $\sigma$ affects the performance?
3. In P4, line 146, the authors stated that As $t \rightarrow \infty$, the SDE converges to a stationary state $x_T ∼ N(x_t | \mu, \lambda^2)$. This formula does not make sense to me. Please check if t is T.

---

> ### Author Response · Authors · 2025-11-20
> **Response to Reviewer nRYJ (part 1/2)**
>
> Thank you for your thoughtful review and constructive feedback. Below, we provide our point-to-point response to addresses your concerns:
>
> > Q1: The authors misinterpretation of forward diffusion as backward. This can be observed from they actually started from $x_T \sim p_{data}$ to $x_0 \sim p_{prior}$.
>
> We appreciate the reviewer’s concern. However, we respectfully disagree with the claim that our method misinterprets the forward and backward directions. In the background section, we intentionally write $x_0 \sim p_{\text{prior}}$ and $x_T \sim p_{\text{data}}$ to unify the notation across different generative frameworks. In our FoD, both training and sampling are strictly defined in the forward time direction, i.e., from prior $x_0$ to data $x_T$. Our model directly learns the forward drift term $\mu - x_t$ and applies it to the same forward SDE for data sampling, without requiring reverse-time dynamics or scores which are used in standard diffusion backward processes. This is also fundamentally different from diffusion bridge models such as GOUB, which introduce bridges via Doob's $h$-transform and still rely on the diffusion forward-backward construction. In contrast, FoD requires neither a reverse process nor a bridge constraint, and its entire generative mechanism is governed by a single forward SDE with analytically tractable transitions. We have clarified this point more explicitly in the revised manuscript.
>
> > Q2: "the flow field decays multiplicatively from its initial value"...the variance of the noise can explode.
>
> We would like to clarify that FoD's multiplicative structure does not lead to variance explosion, owing to its mean-reversion property in both the drift and diffusion functions. To make this clear, let us recall our solution to FoD at the terminal state $x_T$:
> $$
> x_T = (x_0 - \mu ) e^{-\int_0^T (\theta_t + \frac{1}{2}\sigma_t^2 ) \mathrm{d}t + \int_0^T{\sigma}_t {\mathrm{d}w_t}} + \mu.
> $$
> Here, the log-normal noise is introduced by the term $e^{\int_0^T{\sigma}_t {\mathrm{d}w_t}}$, which indeed has a large variance due to the integration over all $\sigma$ values. However, we observe that the other integral in this equation, i.e. $\int_0^T (\theta_t + \frac{1}{2}\sigma_t^2 ) \mathrm{d}t$, also can be large due to positive $\theta$ and $\sigma$ schedules. Consequently, the exponential coefficient $e^{-\int_0^T (\theta_t + \frac{1}{2}\sigma_t^2 ) \mathrm{d}t}$ becomes very small (approaching zero), which drives the state $x_T$ towards the target mean $\mu$, regardless of the noise magnitude. This behavior is further illustrated by visualizing the sampling process of FoD on different tasks, which has been added to *Figure 6, Figure 8, and Figure 9* in the revised paper.
>
> > Q3: the authors used extremely small diffusion coefficients $e^{-\int_0^t (\theta_s + \frac{1}{2}\sigma_s^2 ) ds}$, this made it actually more close to deterministic method...the second row images in Figure 1 seemed so deterministic.
>
> As explained in Q2 above, the exponential coefficient $\delta=e^{-\int_0^T (\theta_t + \frac{1}{2}\sigma_t^2 ) \mathrm{d}t}$ naturally becomes small due to the integration over positive $\theta$ and $\sigma$ schedules. However, this does not imply that the diffusion becomes deterministic. Instead, the coefficient value determines only the convergence rate of the data transformation between distributions. To illustrate this, we visualize the sampling process of FoD with different coefficients in *Figure 7* of the revised manuscript. As can be observed, the diffusion volatility remains similar across different $\delta$ values, while processes with smaller coefficients converge more rapidly toward the clean target data. Overall, the sampling trajectories appear temporally shifted (or horizontally scaled) as $\delta$ increases from 0.001 to 0.1.
>
> In addition, for the example shown in Figure 1, we clarify that the inpainting process is not deterministic. Recall the mean-reversion term $x_0 - \mu$ in the solution: for the unmasked (non-degraded) regions, FoD does not introduce additional noise since $x_0$ and $\mu$ share the same pixel values, i.e. $x_0[\text{unmask}] - \mu[\text{unmask}] = 0$, throughout the sampling process. As a result, FoD focuses only on the degraded areas. This behavior varies across tasks, depending on the degree of difference between the source and target images. We provide more visual examples in *Figure 6 (left), Figure 8, and Figure 9* in the revised paper.

---

> ### Author Response · Authors · 2025-11-20
> **Response to Reviewer nRYJ (part 2/2)**
>
> > Q4: PROPOSITION 3.1 seems like an existing result and no need to prove again.
>
> We respectfully clarify that, although the derivation of our solution uses standard stochastic calculus tools (e.g., Itô's formula), the FoD SDE presented in Eq.(6) is, to the best of our knowledge, new and has not been explored in the generative modelling literature. Proposition 3.1 is included to explicitly state the closed-form transition density. We provide its full derivation to make the paper self-contained and to highlight the analytical tractability of our method. Crucially, both our SDE formulation, i.e., the state-dependent diffusion, and its solution differ from existing Brownian motion based (e.g., VE-SDE) and mean-reverting based (e.g., VP-SDE, IR-SDE, GOUB) approaches. If the reviewer is aware of prior work that includes this exact SDE and its solution, we would greatly appreciate pointers to the relevant references and would be happy to acknowledge and cite them in the revised manuscript.
>
> > Q5: Add GOUB visual results to the figures.
>
> We appreciate the reviewer’s suggestion regarding the visual comparisons with GOUB. We have now added more visual comparisons with GOUB on both image restoration and image-to-image translation in *Figure 2, Figure 3, and Appendix C.4* in the revised paper. In addition, we further added the comparison of their sampling processes on different tasks in *Figure 6, Figure 8, and Figure 9*.
>
> > Q6: Discuss more how selection of the coefficients impact the behavior of the diffusion.
>
> As explained in Q2 and Q3, the coefficient $\delta=e^{-\int_0^T (\theta_t + \frac{1}{2}\sigma_t^2 ) \mathrm{d}t}$ determines the convergence rate of the data transformation between distributions. More specifically, a smaller $\delta$ leads to a faster convergence toward the target but does not strongly affect the diffusion volatility. We have added the visualization of the sampling process of FoD with different coefficients in *Figure 7* in our revised paper.
>
> > Q7: Choice of $\theta$ and $\sigma$ affects the performance?
>
> To assess the sensitivity of FoD to the choice of $\theta$ and $\sigma$ schedules, we conduct an ablation experiment in which the FoD is trained with different $\theta, \sigma$ schedule combinations, as summarized in **Table 1** below. Although different schedules produce small variations across metrics, our model consistently outperforms the IR-SDE baseline, indicating that FoD is flexible and robust to different $\theta$ and $\sigma$ schedules. We have added this experiment to the discussion and further provided the training curves in *Figure 6 (right) and Figure 10* in our revised paper.
>
> Table 1. Analysis of the choice of $\theta$ and $\sigma$ schedules.
>
> | Method |  PSNR  | SSIM  | LPIPS  |  FID  |
> |  ----  | ----  |  ----  |  ----  |  ----  |
> | IR-SDE (diffusion baseline)  | 31.65 | 0.904 | 0.047 | 18.64 |
> | linear $\theta$, linear $\sigma$ | 32.48 | 0.918 | 0.045 | 16.11 |
> | cosine $\theta$, linear $\sigma$ | 32.56 | 0.925 | 0.038 | 14.10 |
> | cosine $\theta$, const $\sigma$ | 32.22 | 0.906 | 0.046 | 17.34 |
> | cosine $\theta$, cosine $\sigma$ | 32.73 | 0.931 | 0.039 | 15.12 |
>
>
> > Q8: Check if $t$ is $T$ in line 146.
>
> Thank you for your careful review. We have now corrected the stationary state as $x_T \sim \mathcal{N}(x_T \mid \mu, \lambda^2)$ in the revised manuscript.

---

> > ### Comment · Reviewer_nRYJ · 2025-11-28
> >
> > Thank you for your rebuttal and clarification! Your reply has addressed some of my concerns. Considering the comments from other reviewers, I have decided to raise my score to 4.

---

> > > ### Author Response · Authors · 2025-11-28
> > >
> > > Dear Reviewer nRYJ,
> > >
> > > Thank you for you reply and for taking the time to re-evaluate our work. We appreciate your updated score and your acknowledgment that our rebuttal addressed some of your concerns. In light of our response, could you please let us know what remaining concerns you feel were not fully resolved? We would be happy to provide further clarification and revise our paper accordingly.
> > >
> > > Best regards,
> > >
> > > Authors

---

### Official Review · Reviewer_owVT · 2025-11-01

**Soundness:** 2
**Presentation:** 3
**Contribution:** 2
**Rating:** 4
**Confidence:** 3

**Summary:**

This manuscript proposes a new forward diffusion process for generative modeling. Specifically, the authors leverage a mean-reverting style SDE and apply the mean-reversion terms to both the drift and diffusion coefficients. The designed SDE has a closed-form solution, and the only unknown term is the simulation target. Following the spirit of denoising score matching, the author proposes a simple regression objective to learn it and thus enable simulating the designed SDE. For the sampling option, the author explored both the Euler method and the first-order discretization. Experimental results demonstrate that the proposed technique is applicable on image-to-image tasks.

**Strengths:**

* The writing is easy to follow.
* Using the proposed mean-reverting-style SDE for generative modeling looks novel to me.

**Weaknesses:**

* The claim about "simpler, single" diffusion process is somewhat unconvincing to me. According to the training algorithm and sampling procedure, the effort is almost the same as that of the diffusion models, and the training objective itself needs approximation.
* Another main claim of the paper is that the proposed method can be viewed as a stochastic counterpart to flow matching. However, to me, it would be necessary to compare the established stochastic counterpart, known as diffusion bridges or bridge matching [1, 2], of the flow matching model, both conceptually and empirically (on image-to-image benchmarks).
* For empirical results, it would be better to add Gassuain-to-image generation tasks to demonstrate the effectiveness of the proposed framework.



## References
[1] Peluchetti, Stefano. ‘Non-Denoising Forward-Time Diffusions’. (2023)

[2] Shi, Yuyang, et al. ‘Diffusion Schrödinger Bridge Matching’. (NeurIPS 2023)

**Questions:**

N/A

---

> ### Author Response · Authors · 2025-11-20
> **Response to Reviewer owVT**
>
> We sincerely thank the reviewer for the thoughtful review and for acknowledging the novelty of our mean-reverting-style SDE formulation. Below, we provide additional clarifications and experiments to address the raised concerns:
>
> > Q1: Clarification of the claim: "FoD is a simpler, single diffusion process".
>
> We would like to clarify that the term "simpler" refers to the *conceptual formulation* of our proposed FoD framework, rather than the practical cost of training or sampling. More specifically, traditional diffusion models couple the forward process with a separate reverse-time SDE, which lacks a closed-form solution. This forward-backward construction complicates both the theoretical formulation and the overall understanding of the model, while also requiring a specialized score function for both training and sampling. In contrast, FoD employs a single forward SDE with analytically solvable transitions for data generation, thereby eliminating the need to define or learn a separate reverse-time SDE. We thank the reviewer for raising this concern, and we have clarified this in the introduction of the revised manuscript.
>
> > Q2: Comparison with diffusion bridge models.
>
> We appreciate the reviewer's suggestion regarding comparisons with diffusion bridges. While conceptually related, diffusion bridge models introduce extra theoretical complexity, such as the Doob's $h$-transform[1,4,5], bridge-consistency constraints[2] or the necessity to solve a mixture of diffusion bridges[3], in order to stochastically interpolate between two distributions. In contrast, our framework focuses on deriving and learning a tractable forward-only process without additional constraints. We have included these references and a discussion in the background section of our revised paper.
>
> In addition, as suggested, we implement two state-of-the-art diffusion bridge models, GOUB[4] and UniDB[5], for image-to-image translation tasks (as well as for image restoration, see *Table 1* in the revised paper). The comprehensive evaluation and comparison have been added in *Table 8 and Figure 3* in the revised manuscript. For reference, we provide the comparison of MSE, LPIPS and FID metrics on *edges to handbags* in **Table 1** and report the FID score across all tasks in **Table 2** below. These results show that FoD achieves highly competitive performance across both restoration and translation scenarios, demonstrating a strong generality on conditional image generation.
>
>
> Table 1. Comparison of FoD with other flow matching and diffusion bridge models on the *edges to handbags* translation task.
>
> |  Method  |  MSE  | LPIPS  | FID  |
> |  ----  | ----  |  ----  |  ----  |
> | SDEdit | 0.510 | 0.271 | 26.5 |
> | Rectified flow | 0.088 | 0.241 | 25.3 |
> | GOUB | 0.054 | 0.224 | 9.36 |
> | UniDB | 0.048 | 0.218 | 9.12 |
> | FoD-SDE | 0.025 | 0.198 | 8.45 |
>
> Table 2. Comparison of FID results on four image-to-image translation tasks.
>
> |  Method  |  Edges to handbags  | Labels to facades  | Maps to aerial photos  |  Night to day  |
> |  ----  | ----  |  ----  |  ----  |  ----  |
> | Flow matching | 25.29 | 21.30 | 36.07 | 78.52 |
> | GOUB | 9.36 | 3.47 | 11.62 | 64.35 |
> | UniDB | 9.12 | 2.79 | 11.78 | 64.52 |
> | FoD-SDE | 8.45 | 0.86 | 2.33 | 52.11 |
>
> > Q3: Experiments on Gaussian-to-image generation.
>
> We would like to clarify that we already included this experiment in the original submission as an unconditional image generation task in Appendix C.3. To further demonstrate the effectiveness of FoD in this setting, we have added an ablation study that increases the sampling steps from 100 to 1000, as shown in **Table 3** below. This simple modification improves the FID score and outperforms flow matching with an optimal transport path. These results demonstrate that FoD not only achieves strong performance on image restoration and other image-conditioned generation tasks, but also performs competitively in Gaussian-to-image generation compared to common baselines. We would also like to re-emphasize that the primary aim of this paper is to construct a conceptually simple and effective single diffusion process model, which we expect to be particularly well-suited for image-conditioned generation. This is an active and fundamental direction of generative modelling with broad real-world relevance.
>
> Table 3. Unconditional generation on CIFAR10.
>
> |  Method  |  FID  |
> |  ----  | ----  |
> | Flow matching w/ diff path | 10.31 |
> | Flow matching w/ OT path | 6.96 |
> | FoD-SDE (T=100) | 7.89 |
> | FoD-SDE (T=1000) | 6.59 |
>
> ---
>
> References:
>
> [1] Denoising Diffusion Bridge Models. ICLR 2024.
>
> [2] Diffusion Schrödinger Bridge Matching. NeurIPS 2023.
>
> [3] Non-Denoising Forward-Time Diffusions. Arxiv 2023.
>
> [4] Image restoration through generalized Ornstein-Uhlenbeck bridge. ICML 2024.
>
> [5] UniDB: A Unified Diffusion Bridge Framework via Stochastic Optimal Control. ICML 2025.

---

> > ### Author Response · Authors · 2025-11-28
> > **A Friendly Reminder**
> >
> > Dear Reviewer owVT,
> >
> > This is a friendly reminder that the discussion period is coming to an end. We hope that our rebuttal and the revised paper have addressed all of your concerns, and we would be grateful if you could reassess our work and consider increasing the score.
> >
> > We would also be happy to clarify any remaining questions or issues. Thank you again for your valuable feedback and thoughtful review.
> >
> > Best regards,
> >
> > Authors

---

### Official Review · Reviewer_jcwr · 2025-11-06

**Soundness:** 4
**Presentation:** 4
**Contribution:** 3
**Rating:** 6
**Confidence:** 3

**Summary:**

This paper introduces forward-only diffusion, which replaces the standard diffusion SDE with a mean-reversion term in both the drift and diffusion terms. They derive a tractable solution to the SDE that determines the conditional distribution $p(x_{t+1}|x_t, \mu)$ and parameterize their neural network to learn the flow $\hat{\mu}_{\phi} - x_t$. This enables a tractable loss to minimize the KL between the ground truth conditional and the model's estimate. Experimental results on image restoration tasks are provided, demonstrating improvements over baselines.

**Strengths:**

**(S1)**: Elegant formulation. A single, forward-only SDE for diffusion with the mean-reversion term is a neat formulation for image-restoration and conditional generation tasks. The mean-reversion term enables a state-dependent denoising process that dynamically adjusts to different corruption levels within an image. To me, this makes a lot of sense for conditional generation.

**(S2)**: Tractability and flexibility. The SDE with mean-reversion in the diffusion term still yields a unique, tractable solution and enables a simple loss function that is stable to train. This is a strong point in favor of the method. Providing both Markovian and non-Markovian sampling strategies for this forward-only SDE further improves the flexibility of this approach.

**(S3)**: Good ablations. Nullifying the diffusion term and reverting the SDE back to the flow-matching ODE clearly demonstrates worse performance on image restoration tasks, in terms of structural similarity metrics. This demonstrates the need for stochasticity in the mean-reverting SDE. Another ablation on fast-sampling is helpful.

Overall, I think the idea and application of this paper is novel and interesting, and so I would recommend it for acceptance. If some of my concerns outlined below are addressed, I would be happy to raise my score.

**Weaknesses:**

**(W1)**: Poor unconditional generation. The FID scores on CIFAR-10 (7.89 for FoD-SDE, 5.01 for FoD-ODE) are not competitive with standard forward-backward diffusion models (e.g., Score SDE @ 2.38) or even other forward-only ODE models like Rectified Flow (2.58). While noted as a limitation, this positions FoD as more a specialized method for conditional image generation tasks than as a general generative model.

**(W2)**: Limited exploration of conditional image generation tasks. The paper mainly focuses on image restoration, which is a low-entropy task (i.e. the source is already close to the target image). Some qualitative examples for image-to-image translation are provided, but a more extensive evaluation on translation tasks would be useful to support the generality of this approach. Additional experiments on text-to-image or latent-diffusion architectures would be further welcome for completeness.

**(W3)**: Missing comparisons. Some recent work on diffusion bridges (denoising diffusion bridge models) tackle similar problems as this paper. Comparison and a more detailed discussion around bridge models is missing. While FoD is an instantiation of stochastic interpolant methods, the paper could be strengthened by a more detailed comparison to other recent SI-based methods that have also been applied to image restoration.

**Questions:**

**(Q1)**: Given the primary weakness is unconditional generation, have the authors experimented with modifying the prior distribution to better match the model's log-normal structure (e.g., starting from a log-normal prior)

**(Q2)**: Do the authors have explanations for the behavior between the MC and non-MC samplers in Figure 4? Why are structural metrics better here compared to generation quality metrics? How should one decide between the two samplers?

**(Q3)**: What is $x_s$ and $\mu$ in the tasks outlined in Figure 3? How does the SDE behave when the tasks represents bridging two very semantically different distributions?

**(Q4)**: How sensitive is the model to the choice of the $\sigma_t$ schedule given that it now controls mean-reversion / a state dependent term in the diffusion term?

---

> ### Author Response · Authors · 2025-11-20
> **Response to Reviewer jcwr (part 1/3)**
>
> Thank you for the insightful review and constructive comments, which accurately summarize our paper and our intended contributions. We appreciate your positive feedback and are happy to provide our point-to-point response below:
>
> > Q1: Poor unconditional generation performance.
>
> We agree that, as we noted in the paper, FoD is particularly well-suited and optimized for conditional image generation tasks. For unconditional generation, we reuse exactly the same model setup as the conditional setting, including the architecture, noise schedule, and diffusion steps, etc. This setup partly causes the performance gap between FoD and models specifically tuned for unconditional generation. As an illustrative experiment, we simply increase the diffusion steps of FoD-SDE to 1000, which improves the FID to 6.59, already outperforming flow matching with an optimal-transport path, as shown in **Table 1** below. While this does not yet reach the SOTA performance, it narrows the gap and illustrates a clear upward trend. We expect that further tuning of the architecture and noise schedule could help FoD approach a competitive unconditional generation performance, while preserving its simplicity and strong suitability for conditional generation tasks.
>
> Table 1. Unconditional generation on CIFAR10.
>
> |  Method  |  FID  |
> |  ----  | ----  |
> | Flow matching w/ diff path | 10.31 |
> | Flow matching w/ OT path | 6.96 |
> | FoD-SDE (T=100) | 7.89 |
> | FoD-SDE (T=1000) | 6.59 |
>
> >Q2: Limited exploration of conditional image generation tasks.
>
> Thank you for highlighting the scope of conditional evaluations. Image restoration is widely used as an important preprocessing step for diverse computer vision tasks such as classification and detection.
> Despite involving a relatively "low-entropy" generation process, it remains a fundamental and highly challenging task for most diffusion and flow matching models, especially under severe degradations where the condition only provides partial information.
>
> But we also agree that adding more extensive evaluations on image-to-image translation would strengthen the generality of our approach. As suggested, we implement a flow matching[1] and two diffusion bridge models, GOUB[2] and UniDB[3], on four image-to-image translation tasks and have reported their MSE, LPIPS, and FID metrics in *Table 8* of the revised manuscript. A qualitative comparison is also added in *Figure 3* of the revised manuscript. For reference, we provide the comparison on *edges to handbags* and the FID on all datasets in **Table 2 and Table 3** below. These results show that FoD achieves highly competitive performance across both restoration and translation scenarios, demonstrating a strong generality on conditional image generation. Other suggested experiments such as text-to-image generation and latent-diffusion are promising, and we will be happy to extend FoD on these directions in future work.
>
>
> Table 2. Comparison of FoD with other flow matching and diffusion bridge models on the *edges to handbags* translation task.
>
> |  Method  |  MSE  | LPIPS  | FID  |
> |  ----  | ----  |  ----  |  ----  |
> | SDEdit | 0.510 | 0.271 | 26.5 |
> | Rectified flow | 0.088 | 0.241 | 25.3 |
> | GOUB | 0.054 | 0.224 | 9.36 |
> | UniDB | 0.048 | 0.218 | 9.12 |
> | FoD-SDE | 0.025 | 0.198 | 8.45 |
>
> Table 3. Comparison of FID results on four image-to-image translation tasks.
>
> |  Method  |  Edges to handbags  | Labels to facades  | Maps to aerial photos  |  Night to day  |
> |  ----  | ----  |  ----  |  ----  |  ----  |
> | Flow matching [1] | 25.29 | 21.30 | 36.07 | 78.52 |
> | GOUB | 9.36 | 3.47 | 11.62 | 64.35 |
> | UniDB | 9.12 | 2.79 | 11.78 | 64.52 |
> | FoD-SDE | 8.45 | 0.86 | 2.33 | 52.11 |
>
> ---
> References:
>
> [1] Flow matching for generative modeling. ICLR 2023.
>
> [2] Image restoration through generalized Ornstein-Uhlenbeck bridge. ICML 2024.
>
> [3] UniDB: A Unified Diffusion Bridge Framework via Stochastic Optimal Control. ICML 2025.

---

> > ### Author Response · Authors · 2025-11-20
> > **Response to Reviewer jcwr (part 2/3)**
> >
> > > Q3: Missing comparison and discussion with recent diffusion bridge and SI-based models.
> >
> > Diffusion bridge models aim to stochastically interpolate between two distributions and are also applicable to both image restoration and translation tasks. However, diffusion bridge models introduce an extra bridge term (via Doob's $h$-transform) and still use the coupled forward-backward framework for data sampling, which conceptually complicates the learning and inference. In contrast, FoD enables a simple, single forward process for data sampling under the analytically solvable, mean-reverting SDE framework. We appreciate your suggestion and have added the diffusion bridges to the background section in the revised manuscript. Moreover, we further compared the iterative sampling processes between FoD and diffusion bridges in the discussion (*left of Figure 6*).
> >
> > In addition, we would like to kindly clarify that our original submission already included a comparison with GOUB[2], a representative diffusion bridge model for image restoration. To strengthen the comparison, we implement GOUB also for image-to-image translation and further add a new state-of-the-art diffusion bridge model called UniDB[3] to all experiments, as shown in **Table 2, Table 3** above (for image-to-image translation) and in **Table 4** below (for image restoration). For the SI-based approaches, to the best of our knowledge, there is unfortunately no publicly available implementation for image restoration or translation. Instead, we provide extensive comparisons with the major families of methods relevant to SI, including diffusion models, diffusion bridges, and flow matching, for both image restoration and translation tasks in the revised manuscript.
> >
> >
> > Table 4. Comparison of FoD with other diffusion bridge models (GOUB and UniDB) on the image restoration tasks.
> >
> >
> > (a) Image deraining
> >
> > |  Method  |  PSNR  | SSIM  | LPIPS  |  FID  |
> > |  ---  | ---  |  ---  |  ---  |  ---  |
> > | IR-SDE | 31.65 | 0.904 | 0.047 | 18.64 |
> > | GOUB | 31.96 | 0.903 | 0.046 | 18.14 |
> > | UniDB | 32.05 | 0.904 | 0.045 | 17.65 |
> > | FoD-SDE | 32.56 | 0.925 | 0.038 | 14.10 |
> >
> > (b) Image Low-light enhancement
> >
> > |  Method |  PSNR  | SSIM  | LPIPS  |  FID  |
> > |  ----  | ----  |  ----  |  ----  |  ----  |
> > | IR-SDE | 20.45 | 0.787 | 0.129 | 47.28 |
> > | GOUB | 19.29 | 0.775 | 0.148 | 50.44 |
> > | UniDB | 20.18 | 0.796 | 0.128 | 45.61 |
> > | FoD-SDE | 21.61 | 0.819 | 0.105 | 41.31 |
> >
> > (c) Image dehazing
> >
> > |  Method  |  PSNR  | SSIM  | LPIPS  |  FID  |
> > |  ----  | ----  |  ----  |  ----  |  ----  |
> > | IR-SDE | 25.25 | 0.906 | 0.060 | 8.33 |
> > | GOUB | 25.31 | 0.908 | 0.048 | 8.33 |
> > | UniDB | 25.65 | 0.896 | 0.051 | 8.21 |
> > | FoD-SDE | 26.57 | 0.932 | 0.033 | 8.14 |
> >
> > (d) Image inpainting
> >
> > | Method |  PSNR  | SSIM  | LPIPS  |  FID  |
> > |  ----  | ----  |  ----  |  ----  |  ----  |
> > | IR-SDE | 29.83 | 0.904 | 0.045 | 26.30 |
> > | GOUB | 29.81 | 0.916 | 0.039 | 23.39 |
> > | UniDB | 30.01 | 0.917 | 0.038 | 23.16 |
> > | FoD-SDE | 30.28 | 0.923 | 0.029 | 16.12 |
> >
> >
> > > Q4: Experiments of FoD with different prior distributions (e.g., log-normal prior).
> >
> > We thank the reviewer for this insightful comment regarding alternative priors for unconditional generation. As suggested, we replace the prior from standard normal to log-normal distribution to better match the multiplicative structure of the forward process. Interestingly, this change leads to a significant performance drop (FID increased from 7.89 to 16.45) as shown in **Table 5** below. We believe this is caused by the mismatch between image state space and log-normal state space. Specifically, all states $x_{0:T}$ in our formulation can contain both positive and negative values, whereas a log-normal prior only assigns positive values to $x_0$, leading to a mismatch among states and therefore distorting the training. This ablation experiment suggests that a standard normal prior remains reasonable and suitable for unconditional generation despite FoD's multiplicative structure. We do however still believe that there are other interesting directions to explore to improve the unconditional generation capabilities, such as log-space transformations and optimal transport-based drift paths, which we plan to investigate in future work.
> >
> > Table 5. FoD-SDE with different priors for unconditional image generation.
> >
> > |  Method  |  FID  |
> > |  ----  | ----  |
> > | FoD-SDE + log-normal prior | 16.45 |
> > | FoD-SDE + normal prior | 7.89 |
> >
> > ---

---

> ### Author Response · Authors · 2025-11-20
> **Response to Reviewer jcwr (part 3/3)**
>
> > Q5: Explanations for the behavior between the MC and non-MC samplers in Figure 4.
>
> The different behaviors of FoD with MC and non-MC samplers mainly stem from how the noise is injected into the sampling process. For the MC sampler, the forward transition with log-normal noise is applied to the current state recursively. The prediction error thus accumulates over time, which tends to distort spatial structure (lower SSIM) but increase sample diversity and quality (better FID). In contrast, given an estimated $\mu$, the non-MC sampler applies the transition on the initial state $x_0$ without recursive updates, which naturally preserves the structure but also reduces sample diversity. In practice, the MC sampler is preferable for highly ill-posed problems (e.g., image-to-image translation) while the non-MC sampler is more suitable for structure-preserving tasks such as image restoration. We have added this explanation to the revised version.
>
> > Q6: What is $x_s$ and $\mu$ in the tasks outlined in Figure 3?
>
> For image-to-image translation tasks in Figure 3, the initial state $x_0$ and the mean term $\mu$ are source (e.g., edges and labels) and target (e.g., handbags and facades) images, respectively. And any intermediate state $x_s$ is the stochastic interpolation between $x_0$ and $\mu$ following the solution of FoD. When bridging two semantically different distributions, FoD produces a smooth and stable trajectory from $x_0$ toward $\mu$. To make this clear, we have added illustrations of the FoD sampling process and compared it with diffusion bridges in *Figure 6, Figure 8, and Figure 9* in the revised manuscript.
>
> > Q7: How sensitive is the model to the choice of the $\sigma_t$ schedule.
>
> To assess the sensitivity of FoD to the choice of $\theta$ and $\sigma$ schedules, we conduct an ablation experiment in which FoD is trained with different $\theta, \sigma$ schedule combinations, as summarized in **Table 6** below. Although different schedules produce small variations across metrics, our model consistently outperforms the IR-SDE baseline, indicating that FoD is flexible and robust to different $\theta$ and $\sigma$ schedules. We have added this experiment to the discussion and further provided the training curves in the right of *Figure 6* and *Figure 10* in our revised paper.
>
> Table 6. Analysis of the choice of $\theta$ and $\sigma$ schedules.
>
> | Method |  PSNR  | SSIM  | LPIPS  |  FID  |
> |  ----  | ----  |  ----  |  ----  |  ----  |
> | IR-SDE (diffusion baseline)  | 31.65 | 0.904 | 0.047 | 18.64 |
> | linear $\theta$, linear $\sigma$ | 32.48 | 0.918 | 0.045 | 16.11 |
> | cosine $\theta$, const $\sigma$ | 32.22 | 0.906 | 0.046 | 17.34 |
> | cosine $\theta$, linear $\sigma$ | 32.56 | 0.925 | 0.038 | 14.10 |
> | cosine $\theta$, cosine $\sigma$ | 32.73 | 0.931 | 0.039 | 15.12 |

---

> > ### Comment · Reviewer_jcwr · 2025-11-27
> >
> > Thank you for the response. My main concerns are addressed, and I am happy to increase my score. Good work!

---

> > > ### Author Response · Authors · 2025-11-27
> > >
> > > Thank you so much for taking the time to reconsider and increase the score! We’re glad to hear that our response and the revision addressed your concerns, and we sincerely appreciate your thoughtful engagement with our work.

---

### Author Response · Authors · 2025-11-20
**Summary of changes in revision**

We appreciate all valuable feedback and comments from the reviewers. The revised paper has now been uploaded. In this version, we added several new experiments, including **extensive evaluations on image-to-image translation**, **comparisons with diffusion bridges**, **schedule sensitivity** and **log-normal prior** ablation studies, and additional **visualizations of the FoD sampling process**. A more detailed summary of updates is provided below for the convenience of all reviewers and the AC.

1. Added more discussion about diffusion bridges in Section 2.1. Added the comparison and discussion of the sampling process between FoD and diffusion bridges in *Figure 6 (left), Figure 8, and Figure 9*.
2. Updated *Table 1* with a diffusion bridge baseline, UniDB[1], which is a state-of-the-art method for diffusion-based image restoration.
3. Added *Table 3* and *Table 8* for an extensive evaluation on image-to-image translation benchmarks (with MSE, LPIPS, and FID metrics). The comparisons include a flow matching method and two diffusion bridge models. A qualitative comparison is also added in *Figure 3*.
4. Added the discussion about the choice of $\theta$ and $\sigma$ schedules in Section 5 (*right of Figure 6*) and in Appendix C.2 (*Figure 10 and Table 6*).
5. Added *Figure 7* in Appendix C.1 to analyze the effect of different time interval coefficient values.
6. Added *Table 7* to analyze different priors (normal and log-normal) for unconditional generation.
7. We have further revised the paper to include all reviewer comments and improve the clarity. In addition, we added more experimental discussion and qualitative comparisons in the Appendix.

---

References:

[1] Zhu et al. UniDB: A Unified Diffusion Bridge Framework via Stochastic Optimal Control. ICML 2025.

---

### Author Response · Authors · 2025-11-30
**Summary of Our Rebuttal**

We sincerely thank all reviewers and the area chairs for their insightful feedback and for taking the time to review our paper. We have provided detailed point-to-point responses in each review thread and have substantially enhanced the paper with suggested new experiments and clarifications. For convenience, the table below briefly summarizes the main concerns of each reviewer and our corresponding responses:


|  Reviewer  |  Main concerns  | Our response |
|  ----  | ---- | ---- |
| `jcwr` | (1) Unconditional generation performance. | Clarified the reason and improved the FoD-SDE result (updated *Appendix C.3* and *Figure 12*). |
| | (2) Exploration of conditional generation on high-entropy tasks. | Added extensive evaluations on four image-to-image translation tasks (updated *Figure 3*, added *Table 3* and *Table 8*) |
| | (3) Missing comparisons. | Added comparisons with two diffusion bridge models, GOUB[1] and UniDB[2], on all tasks (updated *Table 1* and *Table 9*, added *Table 3* and *Table 8*).|
| `owVT` | (1) Claim about the "simpler, single" diffusion process. | Clarified that this refers to the conceptual formulation rather than the practical training cost (updated *Section 1*). |
| | (2) Comparison with diffusion bridge models. | Added two diffusion bridge models, GOUB and UniDB, on all tasks. Expanded the background section on diffusion bridge models (updated *Section 2.1*). |
| `nRYJ` | (1) FoD is actually a backward diffusion. | Clarified the notation. Clearly explained that both our training and sampling are strictly defined in the forward time direction. |
| | (2) The setup resembles the GOUB (diffusion bridge). | Clarified the fundamental difference between FoD and diffusion bridge models. Added comparison of their sampling processes (added *Figure 6*, *Figure 8*, and *Figure 9*). |
| | (3) Small coefficient $e^{-\int_0^T (\theta_t + \frac{1}{2}\sigma_t^2 ) \mathrm{d}t}$ makes FoD deterministic. | Explained that the small coefficient value is reasonable given positive $\theta_t$ and $\sigma_t$ schedules, and that it **does not** make FoD deterministic. Visualized the sampling process under different coefficients (added *Figure 7*). |
---
We also added additional clarifications and ablation studies (e.g., on the log-normal prior and the choices of the $\theta_t, \sigma_t$ schedules) to address the remaining concerns and questions. Please find more details in the rebuttal and the revised manuscript.

In general, Reviewer `jcwr` accurately summarized our paper and intended contributions, and Reviewer `owVT` further acknowledged the novelty of our mean-reverting-style SDE formulation. However, we would like to respectfully note that **Reviewer `nRYJ` appears to have a fundamental misunderstanding of our method**, particularly regarding *1) the definition of the forward diffusion process* and *2) the role of the small coefficient $e^{-\int_0^T (\theta_t + \frac{1}{2}\sigma_t^2 ) \mathrm{d}t}$*. This misunderstanding seems to be the main reason for the low score in their original review. We believe that our rebuttal and the revised manuscript have now thoroughly addressed their concerns, both conceptually and empirically, through clear explanations and additional ablation experiments.


---

References:

[1] Image restoration through generalized Ornstein-Uhlenbeck bridge. ICML 2024.

[2] UniDB: A Unified Diffusion Bridge Framework via Stochastic Optimal Control. ICML 2025.

---

### Meta-Review · Area_Chair_saKw · 2026-01-07

**Summary:**

Reviewers agreed the paper has a neat mean-reverting forward SDE formulation and strong results on image restoration / image-conditioned tasks, but the decision hinges on three recurring concerns: (i) the “forward-only / simpler” positioning was questioned as not clearly simpler in practice, (ii) the relationship to diffusion bridges / bridge matching needed clearer conceptual and empirical separation, and (iii) unconditional generation performance was viewed as weak relative to standard diffusion baselines, raising questions about generality beyond conditional settings.

**Reviewer Concerns:**

Addressed in rebuttal: added diffusion-bridge baselines (GOUB/UniDB) and expanded conditional evaluations (e.g., more i2i translation), and clarified notation/forward-time framing; this satisfied at least one positive reviewer.

Still outstanding: unconditional generation remains a stated weakness (improved but still not clearly competitive with standard diffusion), and there is no evidence that the most skeptical reviewer’s concerns about “simplicity” and positioning vs. bridges were fully resolved during discussion.

**Reviewer Scores:**

jcwr: likely 6 → 6/8, since they stated their main concerns were addressed and they were happy to increase their score.

owVT: likely remains 4 → 4 (no follow-up indicating a change of view).

nRYJ: 2 → 4, explicitly stated they decided to raise to 4 but had trouble updating the score in the system.

---

### Decision · Program_Chairs · 2026-01-26

Reject